# Territoriality modulates the coevolution of cooperative breeding and female song in songbirds

Kate T. Snyder [1,2], Aleyna Loughran-Pierce[1,3] & Nicole Creanza [1,2] ✉

Birdsong has historically been characterized as a sexually selected, primarily male behaviour. Recent findings suggest that female song is widespread, raising questions about how social functions of birdsong shape song evolution. Certain social behaviours, such as cooperative breeding, could alter selection pressures on both sexes and potentially influence the evolution of both female and male song. Here we use phylogenetic comparative analyses across 1,041 songbird species to examine relationships between cooperative breeding, female song and male song characteristics. We show robust bidirectional coevolutionary dynamics between cooperative breeding and female song that persist when controlling for territoriality, allometry, phylogenetic uncertainty, geographical sampling and analytical biases. While cooperative breeding and female song commonly co-occur in strongly territorial systems, their association is especially pronounced in weakly territorial systems, where they co-occur much more often than expected by chance. Additionally, we observe that male song repertoire size evolves more slowly in cooperative breeding lineages. These findings demonstrate that cooperative breeding shapes the evolution of vocal communication differently based on territorial context and sex, with female song potentially serving crucial but understudied functions related to social cohesion in cooperative systems, particularly in species where territorial conflict is reduced.

Song is traditionally viewed as a sexually selected male behaviour with functions related to mate attraction and territory defence[1], but increased recognition that female songbirds sing has revealed that this framing may be incomplete[2,3]. Data on the biological contexts of female song are sparse for most species; however, broadly, several patterns of singing behaviour have been observed, including male–female duetting and agonistic female–female interactions as well as within-pair or within-group interactions[3]. These behavioural patterns correspond with several hypothesized functions of female song: mate attraction, intrasexual competition, territory defence, pair-bond maintenance and social cohesion within groups[4,5]. A recent quantitative review reinforced that territory defence, intrasexual competition and intrapair communication are the most commonly supported functions of female song, while its use in mate attraction is rare[6]. Thus, the evolution of song may be subject to evolutionary pressures related to social interaction in addition to traditionally studied sexual selection, a phenomenon that might be especially relevant in females[7,8].

These varied functions of female song are shaped by ecological and life history factors. In particular, certain breeding systems, such as cooperative breeding, change social dynamics and selection pressures

[1]Department of Biological Sciences, Vanderbilt University, Nashville, TN, USA. [2]Evolutionary Studies Initiative, Vanderbilt University, Nashville, TN, USA. [3]School for Science and Math at Vanderbilt, Vanderbilt University, Nashville, TN, USA. ✉e-mail: nicole.creanza@vanderbilt.edu

on both males and females and provide important contexts for studying interactions between perturbed selection pressures and song evolution. Emerging evidence continues to highlight the social complexity of cooperative breeding systems, with recent work documenting cryptic reciprocal helping relationships between kin and non-kin that persist across breeding seasons[9]. Such durable social affiliations probably require sophisticated communication systems for coordinating helping behaviours, recognizing group members and maintaining social bonds; these communication systems might be more valuable to a group with shared resources to defend or easier to maintain when the group has a defined territory. Therefore, understanding how cooperation and territoriality interact to shape vocal evolution is important for explaining the diversity of singing behaviours across species.

Cooperative breeding, which is broadly defined as a breeding system in which non-parental adults assist parents in rearing their young, has evolved many times in the avian lineage and is observed to different extents in ~13% of Oscine songbird species (Passeri)[10]. In cooperative systems, sexual selection pressures on females are predicted to increase through intensified intrasexual competition for breeding status and associated increase in variance of reproductive success among females[11,12], although the extent of this reproductive skew can vary widely according to species, especially in avian taxa[13]. The effects of cooperative breeding on sexual selection pressures in males are even less straightforward[14,15]. Cooperative breeding often arises from kin-based social structures, with both inclusive and direct fitness benefits for group members, such as resistance to predation through increased group size[16–19]. Notably, cooperative breeding often co-occurs with territoriality and stable social bonds[20]. Therefore, the fitness benefits of cooperative breeding are somewhat entwined with other forms of group living. In songbirds, how these complex social systems affect the functions of and selection pressures on male and female song remains poorly understood. However, emerging evidence suggests that there may be important evolutionary interactions between social stability and vocalizations[21]; for example, cooperative breeding appears to promote the evolution of larger repertoires of functional vocalizations[22]. In addition, several traits related to social network stability, including territoriality, monogamy and long-term pair bonds, are correlated with the presence of female song in several songbird clades[23]. Further, recent phylogenetic analyses[24] found that year-round territoriality and biparental care are the strongest predictors of female song incidence across songbirds, suggesting that ecological factors related to resource defence and parental investment shape vocal evolution. Notably, this study found an additional, albeit weaker, persistent link between cooperative breeding and female song, despite cooperative breeding being considered a subset of biparental care. Overall, there are hints that living in social groups, defending territories and engaging in cooperative caretaking might influence the evolution of female and male vocalizations, but the intricacies of these links have not been fully disentangled.

In this study, we compiled a dataset of Oscine species that includes species-level cooperative breeding classification, data on female song presence, several features of (putatively male) song complexity and performance and new classifications of territorial intensity. Using phylogenetically informed analyses across 1,041 species, we tested whether cooperative breeding influences the evolution of male song traits or female song presence, and how territoriality shapes those interactions. We also evaluated other social variables to compare the effects of cooperative caretaking with other factors related to group living and social stability. Our analyses reveal that the evolutionary association between cooperative breeding and female song is both robust and context-dependent, with an outsized effect in weakly territorial systems. In contrast, male song repertoire size evolves more slowly in cooperative breeding lineages regardless of territorial context. These sex-specific, context-dependent patterns suggest that multiple evolutionary pathways could promote female song: resource

defence in territorial species and social coordination or cohesion in cooperative contexts.

## Results

In this study, we synthesized four new datasets and analysed them together with large-scale avian phylogenies. First, we updated our previously published dataset on species-level song features encompassing song complexity and performance metrics for 339 species (Supplementary Dataset 1). Second, we combined and reconciled data sources reporting the species-level prevalence of cooperative breeding for 4,373 Oscine species (Supplementary Dataset 2). Third, we combined and updated datasets on the presence or absence of female song, obtaining classifications for 1,094 species (Supplementary Dataset 3). Fourth, we refined existing territorial behaviour classifications[21] by developing new designations of weak or absent territoriality versus strong territoriality for 875 species based on detailed behavioural criteria (Supplementary Dataset 4). We also compiled species-level data[16,25] on other forms of social organization (Supplementary Tables 1–2). Together, these datasets allowed us to elucidate the complex relationships between territorial behaviour, cooperative breeding and the evolution of song in both sexes.

### Song feature evolution in the context of cooperative breeding and other sociality characteristics

We found that none of the species-level song features were significantly different between non-cooperative and cooperative breeding species (phylogenetic analysis of variance (phylANOVA) $P > 0.05$; Supplementary Table 3). However, we found that the rates of evolution of two song metrics associated with song complexity, syllable repertoire size and song repertoire size, may have differed between non-cooperative and cooperative lineages, with both evolving more rapidly in non-cooperative lineages in over 95% of simulations, although this difference was only significant for song repertoire size (Brownie likelihood ratio test: song repertoire $P = 0.024$, $n = 209$ species; syllable repertoire $P = 0.136$, $n = 118$ species; Fig. 1 and Supplementary Table 4). We obtained qualitatively similar results for song repertoire evolution when using variations of the cooperative breeding or song repertoire data across cited sources, when using variations of the phylogeny to account for phylogenetic uncertainty, or when performing a jackknife analysis by removing each family in turn (Extended Data Fig. 1 and Supplementary Tables 4 and 5).

Similarly, we found no significant song repertoire size differences between species in different states of any of the discrete sociality traits (phylANOVA $P > 0.05$; Supplementary Table 3). However, in our analyses of evolutionary rates, we found that song repertoire probably evolved faster in lineages with non-familial versus familial living (Brownie likelihood ratio test $P = 0.0042$, $n = 156$ species; Supplementary Table 4), faster in lineages with social bonds lasting one breeding season or less ($P = 0.0079$, $n = 129$ species) and fastest in pair-based social systems than any other social grouping mode ($P = 0.0004$, $n = 127$ species; Extended Data Fig. 2a and Supplementary Table 6a). We also found that song repertoire size evolved fastest in lineages with weak, seasonal or absent territoriality (territory: binary $P = 0.011$, three-level $P = 0.012$, $n = 216$ species; Extended Data Fig. 2b and Supplementary Tables 4 and 6b). The analysis comparing song repertoire evolution between lineages with each combination of cooperative breeding and binary territoriality states (non-cooperative breeding with weak territoriality, non-cooperative breeding with strong territoriality, cooperative breeding with weak territoriality and cooperative breeding with strong territoriality) showed that song repertoire evolved more slowly in both cooperative states than in either non-cooperative state, but that non-cooperative weakly territorial lineages had faster song repertoire evolution than non-cooperative strongly territorial lineages ($P = 0.011$, $n = 152$ species; Extended Data Fig. 2c and Supplementary Table 6c). In ref. [16],

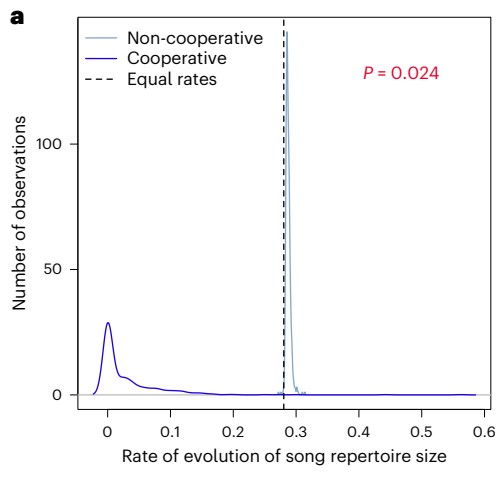
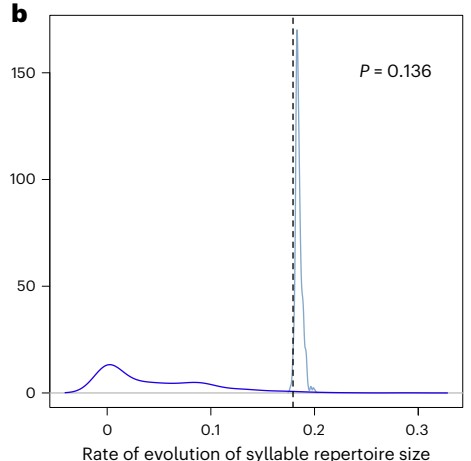

**Fig. 1 | Testing whether the rate of evolution of song repertoire and syllable repertoire (both natural-log-transformed) differs based on cooperative breeding. a**, Song repertoire evolved faster in non-cooperative breeding lineages in 99.2% of simulations, with a Brownie likelihood ratio test *P* = 0.024 (one-sided, d.f. = 1, *n* = 209 species). **b**, Syllable repertoire showed a similar trend of faster evolution in non-cooperative breeding lineages in 98.8% of simulations of the Brownie algorithm[71] (Brownie likelihood ratio test *P* = 0.136, one-sided, d.f. = 1, *n* = 118 species). Both analyses were run for 500 simulations.

social system data for 1,203 species were classified into either three or four states; our analyses showed that song repertoire evolved more rapidly in non-familial, non-cooperative lineages than in lineages with familial living plus cooperative breeding, familial living without cooperative breeding or non-familial living plus cooperative breeding (Brownie likelihood ratio test: three-state *P* = 0.002, four-state *P* = 0.001, *n* = 156 species; Extended Data Fig. 2d,e and Supplementary Table 6d,e), recapitulating our findings from analysing the two traits separately, namely that song repertoire size evolves more slowly both in familial lineages and in cooperative lineages (Supplementary Table 4).

### Female song and cooperative breeding

We were able to obtain species-level classifications of both female song presence and cooperative breeding for 1,041 Oscine species, and compared their evolution using the assembled species-level data as character states at the tips of a published avian phylogeny[26] (Fig. 2). By simulating the evolution of these traits[27], we found evidence to reject the null hypothesis that female song and cooperative breeding evolve independently (empirical *P* = 0.002), a finding primarily driven by a significantly higher-than-expected co-occurrence of female song presence and cooperative breeding (Fig. 3a and Extended Data Table 1). This trend in ancestral state association persisted when we iteratively removed taxonomic families (Supplementary Table 7), and when we used a set of randomly sampled phylogenies, a consensus phylogeny calculated using an alternate method and alternative cooperative breeding classification methods (Extended Data Fig. 3 and Supplementary Tables 8 and 9).

By independently simulating transitions between breeding systems and female song states, we teased out some potential directionality in the evolutionary patterns underlying this relationship. We found that transitions from presence to absence of female song were less frequent in cooperative breeding lineages than expected, suggesting that cooperative breeding might slow the loss of female song (Fig. 3b). In contrast, transitions from presence to absence of female song were more frequent than expected in non-cooperative breeding lineages. These trends were consistent when we ran this analysis using alternate data or phylogenies (Extended Data Fig. 4). Additionally, the presence of female song may promote a transition to cooperative breeding (Fig. 3b), although this trend was not as robust to alternative cooperative breeding classification methods (Extended Data Fig. 4e–i).

### Accounting for other social variables

Several social traits co-occurred with cooperative breeding more often than expected in evolutionary history: social living in groups smaller than about 30 individuals, social bonds longer than one breeding season, familial living, acolonial living, social monogamy, year-round and strong territoriality, and having more than two caretakers of young (Supplementary Table 10). Despite the high degree of co-occurrence between familial living and cooperative breeding, familial living and female song did not co-occur significantly more than expected (*P* = 0.092, *n* = 468 species; Extended Data Fig. 5a and Extended Data Table 1). A similar analysis using the four-state breeding system classification dataset showed that female song occurred significantly more often than expected in systems with kin-cooperative breeding and significantly less often in systems with non-kin-cooperative or non-familial, non-cooperative systems (Extended Data Fig. 5b and Supplementary Table 11).

The only social variable that significantly co-occurred with both cooperative breeding and female song was territoriality, categorized by either territory strength or year-round territoriality (Supplementary Table 10 and Extended Data Table 1). However, an initial assessment of species counts suggested that this mutual association with territoriality only partially explained the link between female song and cooperative breeding (Supplementary Table 12). Thus, we included these territoriality variables in all subsequent analyses to thoroughly test whether the association between female song and cooperative breeding is explained by their mutual association with territoriality. Simulated transition-count analyses split according to territoriality state suggested that cooperative breeding might slow the loss of female song primarily in strong territoriality, and that non-cooperative breeding might accelerate the loss of female song primarily in weak territoriality (Fig. 3c,d).

To further probe the interactions between female song, cooperative breeding and territoriality, we also performed phylogenetic path analyses using networks that included these three traits, and species body mass (log-transformed) to account for allometric scaling effects (Supplementary Fig. 1). We found that the models with the highest support (delta C-statistic information criterion corrected for small sample sizes (ΔCICc) < 2) always included a link between cooperative breeding and female song, although the directionality of that link could not be resolved; we found evidence to support both female song positively influencing cooperative breeding and cooperative breeding positively influencing female song, suggesting the

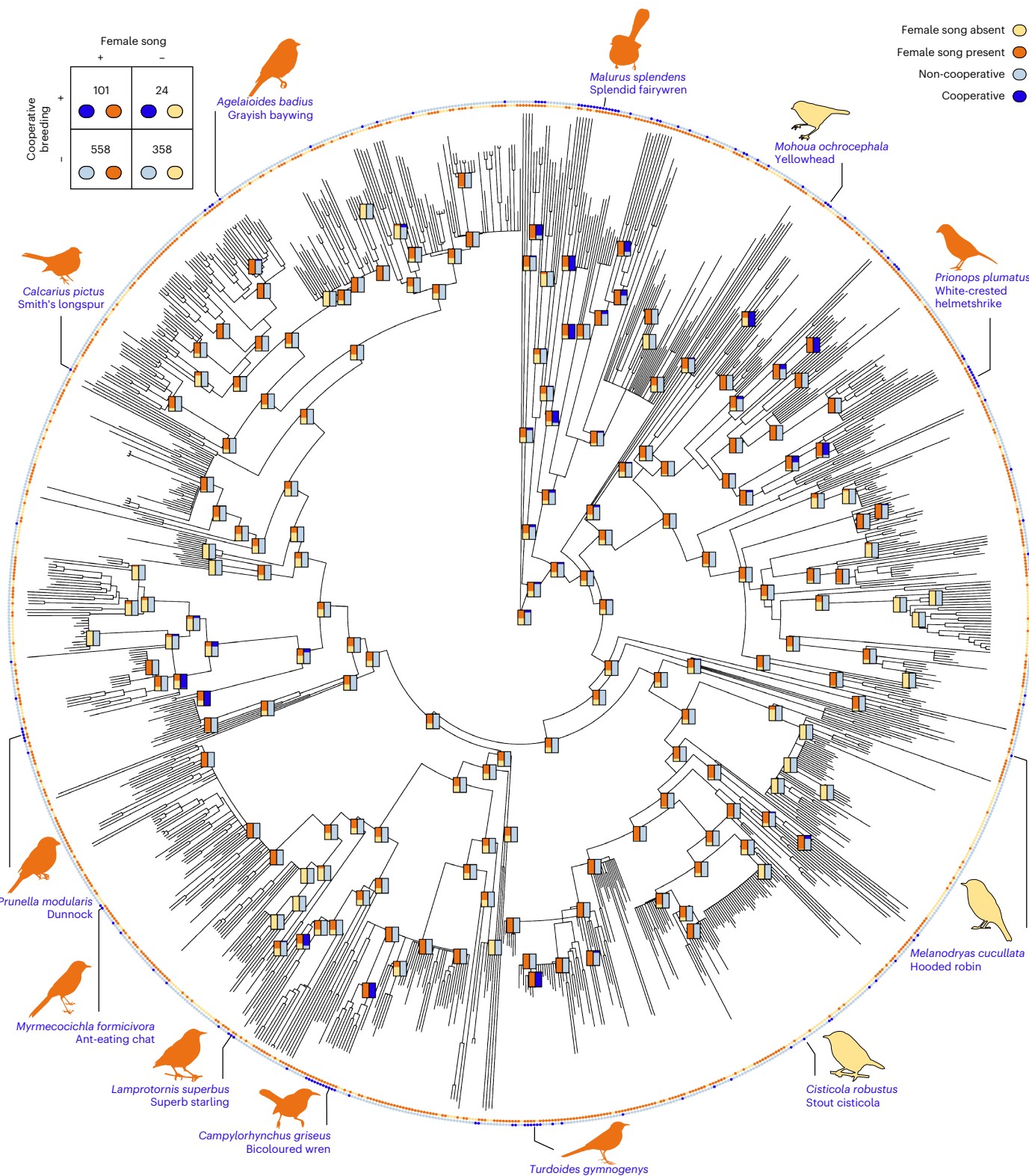

**Fig. 2 | Phylogeny showing the classifications of cooperative breeding (outer circle) and female song (inner circle) for 1,041 Oscine species.** Top left, The table shows the number of species in each category, with female song overrepresented in cooperative breeding species. As landmarks, we illustrate several well-studied species that breed cooperatively, with their silhouettes coloured according to whether female song is present in the species. We note that there are multiple evolutionary gains and losses of both cooperative breeding and female song. In general, the ancestral nodes that are estimated to lack cooperative breeding (light blue) tend to also lack female song (yellow), and the ancestral nodes that are estimated to have cooperative breeding (dark blue) also usually have female song (orange). Credit: Bird silhouettes from Phylopic (http://phylopic.org/) under a Creative Commons license: *Molothrus ater*, *Turdus migratorius*, *Campylorhynchus rufinucha*, *Passer domesticus* and *Zonotrichia leucophrys*, Andy Wilson (CC0 1.0); *Malurus cyaneus*, Michael Scroggie (CC0 1.0); *Mohoua albicilla*, Amy Whitehead (CC0 1.0); *Malaconotidae* and *Cisticolidae*, Lucy_the_bob_man (PDM 1.0); *Petroicidae*, Ferran Sayol (CC0 1.0); *Sturnus vulgaris*, Oier (PDM 1.0); *Catharus ustulatus*, Sharon Wegner-Larsen (CC0 1.0).

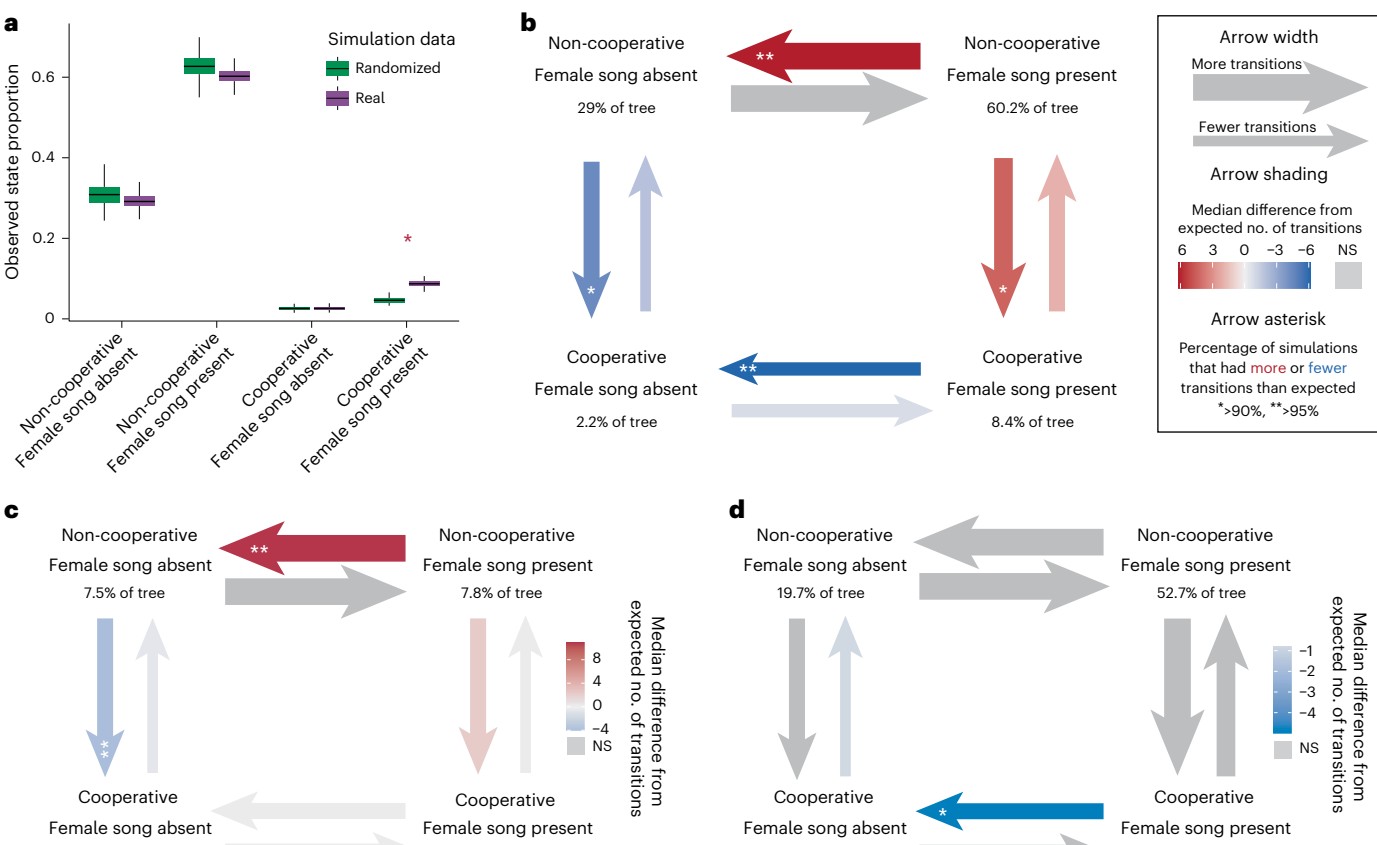

**Fig. 3 | Coevolution and correlated evolution of female song presence and cooperative breeding. a**, Overlap of stochastic character map simulations of female song and cooperative breeding, measured as the proportion of the tree occupied by each state combination (empirical $P = 0.002$, two-sided, based on 500 simulations of stochastic character maps using real or randomized data, $n = 1,041$ species). The box plots show the median (centre line), the 25th and 75th percentiles (box bounds), and whiskers extending to the smallest and largest values within 1.5 times the interquartile range from the lower and upper quartiles; values outside this range (outliers), if present, are not displayed. **b**, Transition likelihoods between female song and cooperative breeding based on the median difference from expected transition counts from 500 simulations. The arrow weights scale to the log-transformed median relative number of state switches that occurred on that topology. The grey arrows indicate rates for

which the observed number of transitions were not significantly different from the expected number, based on an ANOVA. Otherwise, the colour denotes the median difference from the expected number of transitions across simulations, with red indicating that the observed number of state switches was greater than expected and blue indicating fewer state switches than expected. The asterisks indicate that in 90–95% (*) or >95% (**) of simulations, the number of transitions was greater (red arrows) or less (blue arrows) than the calculated expected number of transitions for the simulation in question. The percentage listed under each pair of states is the median percentage of the tree that is in that state combination across all simulations (and thus the values do not necessarily add to 100). **c**,**d**, Numbers of transitions between female song states and cooperative breeding states that occurred in lineages with weak (**c**) and strong (**d**) territoriality. The legend in **b** applies, qualitatively, to **c** and **d**.

potential for a bidirectional coevolutionary relationship (Fig. 4a). This link was robust to data availability bias corrections (Fig. 4b–d and Supplementary Tables 13 and 14), and alternative cooperative breeding classification schemes (Extended Data Fig. 6). When using year-round territoriality instead of strength of territoriality classification, only the positive directional effect of cooperative breeding on female song remained robust (Extended Data Fig. 7).

To estimate the size of the effect of cooperative breeding on female song, and vice versa, in the context of territoriality and body size, we performed phylogenetic generalized linear model analyses. In the best-supported model predicting female song presence, cooperative breeding increased the odds of female song by 5.70-fold (95% confidence interval (CI): 2.17–9.28, $P = 0.004$; Table 1) in species with weak or absent territoriality, a trend that was consistent across alternative cooperative breeding classifications (Supplementary Table 15). However, a significant negative interaction term (odds ratio (OR) = 0.24, CI: 0.16–0.67, $P = 0.024$; Table 1) indicated that this effect was substantially reduced in strongly territorial species. When examining the inverse relationship, female song presence increased the odds of cooperative breeding by 6.88-fold (CI: 3.61–11.28, $P < 0.002$;

Table 1, and Supplementary Table 16) in weakly territorial species, but this effect was also much less pronounced in strongly territorial species (interaction OR = 0.15, CI: 0.09–0.30, $P < 0.002$; Table 1 and Supplementary Table 16). When using year-round territoriality as an alternative binary territoriality classification, the best-supported models included the associations between cooperative breeding and female song, although the estimated effect sizes were considerably weaker (cooperative breeding predicting female song: OR = 1.41; female song predicting cooperative breeding: OR = 1.44; Supplementary Table 17) and did not include interaction terms between cooperative breeding and female song and territoriality. Importantly, the classification based on whether territories are held year-round may conflate species with fundamentally different territorial strategies, for example, by grouping strongly territorial seasonal breeders with non-territorial and weakly territorial species.

**Alternative social predictors of female song**

To determine whether any social variables could better explain female song patterns than cooperative breeding, we used phylogenetic generalized linear models to compare each social variable

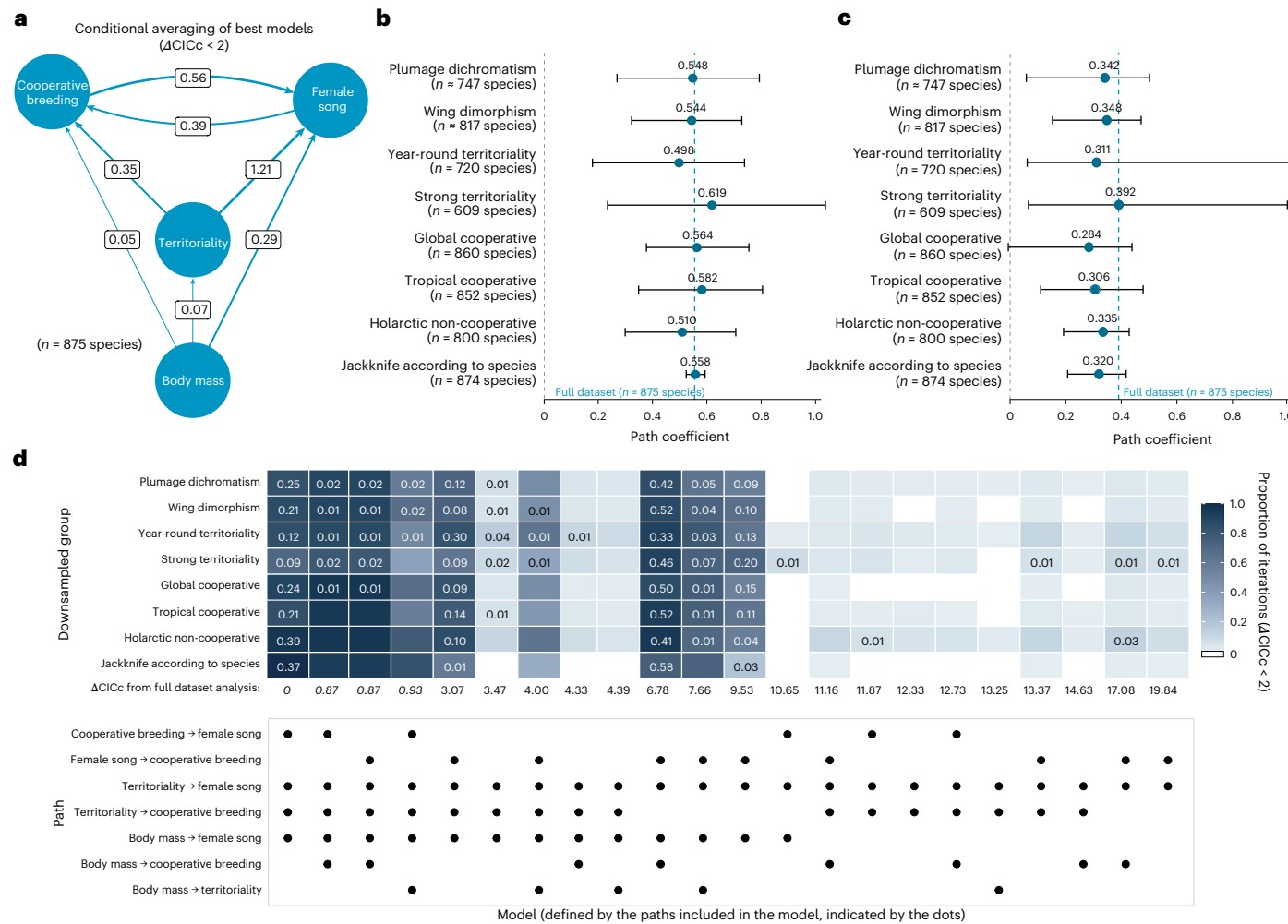

**Fig. 4 | The links between cooperative breeding and female song in phylogenetic path analyses are robust to correcting for data availability biases. a**, Directed acyclic graph showing conditional-averaged path coefficients from phylogenetic path analysis of the relationships between female song presence, cooperative breeding, strength of territoriality and body mass (log-scaled and normalized). The arrow thickness indicates the magnitude of path coefficients, with all paths shown representing positive relationships. Only the four models considered statistically equivalent to the best model (ΔCICc < 2) were averaged. **b**, Forest plot showing the mean and 95% CI for the cooperative breeding → female song path coefficient across 500 iterations for each bias correction type, compared to the value from the full dataset (dashed line). **c**, Forest plot showing the mean and 95% CI for the female song → cooperative breeding path coefficient across 500 iterations for each bias correction type, compared to the value from the full dataset. **d**, Model consistency across bias corrections. Heatmap shows the proportion of iterations in which each phylogenetic path model was supported (had a ΔCICc < 2) across 500 iterations for each bias correction type. Cells for models that were found to be the best-supported model in at least 1% of downsampled simulation iterations are labeled with the proportion of iterations in which the model was the best-supported. The row underneath the heatmap contains the ΔCICc values for each model in the phylopath analysis performed using the full dataset, with the ΔCICc values of the four models that were statistically equivalent (ΔCICc < 2) in the full dataset in italics. Bottom, Dots indicate the presence or absence of each rate in the model. The models shown are all those that were in the top 20 models (of 47 models in total; Supplementary Fig. 1) according to the number of iterations with a ΔCICc < 2 for any of the bias correction types.

against cooperative breeding while controlling for territoriality and body mass. With the caveat that our sample sizes are twofold to four-fold smaller than in Table 1 because we had to exclude species missing data for either variable, we found that cooperative breeding had a higher OR than all social variables, except the average number of caretakers of young; in other words, when we include information on the number of individuals involved in cooperative caretaking instead of just the existence of cooperative caretaking, the model's ability to predict female song presence is even stronger (Table 2 and Supplementary Table 18). Including traits related to social stability, namely longer social bonds, familial living and smaller group size, produced models that are statistically equivalent (delta Akaike Information Criterion (ΔAIC) < 2) to the models with cooperative breeding, whereas including colonial living reduced the model's ability to predict female song (Table 2).

## Discussion

Our phylogenetic comparative analyses across the songbird lineage reveal a significant, bidirectional association between cooperative breeding and the presence of female song that cannot be explained by their shared associations with territoriality, body mass or geographical sampling biases and is robust across several analytical methodologies, alternative phylogenies and alternative methods for cooperative breeding classification. While both traits are indeed more common in territorial species, phylogenetic path analyses and generalized linear models demonstrate an additional evolutionary link between cooperation and female song. Further, we find that this relationship is modulated by territorial context; in strongly territorial species, cooperative breeding and female song are both common and co-occur slightly more often than expected by chance, whereas in species with weak or no territorial defence behaviour, cooperative breeding and female song are overall

**Table 1 | Best base phylogenetic generalized linear models (phyloglm in the R package phylolm), including only territoriality, body mass and female song or cooperative breeding as predictors**

| Predictor (of female song) | OR | CI | P | Predictor (of cooperative breeding) | OR | CI | P |
|---|---|---|---|---|---|---|---|
| Intercept | 0.60 | 0.36–0.94 | **0.028** | Intercept | 0.12 | 0.05–0.26 | **<0.002** |
| Cooperative breeding | 5.70 | 2.17–9.28 | **0.004** | Female song | 6.88 | 3.61–11.28 | **<0.002** |
| Territoriality (weak versus strong) | 3.82 | 2.48–6.59 | **<0.002** | Territoriality (weak versus strong) | 6.05 | 3.07–9.25 | **<0.002** |
| Normalized log(body mass) | 1.45 | 1.14–1.76 | **<0.002** | Female song:territoriality | 0.15 | 0.09–0.30 | **<0.002** |
| Cooperative breeding:territoriality | 0.24 | 0.16–0.67 | **0.024** | | | | |
| Species | 875 | | | Species | 875 | | |
| Model formula: female song=cooperative breeding×territoriality (weak versus strong)+log(mass) | | Model formula: cooperative breeding=female song×territoriality (weak versus strong) | | | | | |

Analyses were performed for 500 bootstrap iterations. *P* values were determined empirically based on the number of bootstrap iterations that fell above or below an OR value of 1.0. Significant *P* values (*P*<0.05) are shown in bold.

less common but co-occur much more often than expected. These dynamics suggest that female song evolution could follow distinct pathways depending on social context; whereas territorial species may maintain female song for resource defence, non-territorial cooperative breeders may have evolved female song through different selective mechanisms, potentially related to the unique social dynamics of cooperative groups. In light of these findings, we consider evidence for and against three hypothesized functions of female song in cooperative breeding systems: sexual competition between females, social conflict and social cohesion.

First, the sexual competition hypothesis predicts that female song in cooperative breeding systems is promoted by intensified intrasexual competition among females because of reproductive skew[28,29]. Under this hypothesis, competition for limited breeding positions, with unsuccessful females serving in helper roles, would amplify sexual selection on females. If song mediates this competition, we would expect female song to co-occur with cooperative breeding. Female song has indeed been linked to mate defence against other females, particularly in polygynous contexts[30,31], suggesting that female song can function in intrasexual competition. Additionally, our analyses show that the evolution of male song repertoire size was slower in cooperative breeding lineages, contrasting with the accelerated song repertoire evolution seen in polygynous systems[32]. This potential evidence for reduced sexual selection pressure on males in cooperative systems could theoretically correspond to increased pressures on females if cooperative breeding reverses typical sex differences in reproductive variance (per Bateman's principle)[15]. However, empirical evidence from cooperative breeding songbirds suggests that helping is male-biased or mixed-sex but not female-biased[33] and female helpers were not typically linked to extreme reproductive skew[13]. Together, these patterns suggest that the conditions necessary for intensified female sexual competition—female-biased helping driven by reproductive monopolization—are not characteristic of cooperative breeding songbirds, casting doubt on the hypothesis that female song evolves in these systems because of reproductive skew.

Second, the social competition hypothesis proposes that female song mediates competition for ecological resources, particularly territories[34]. Under this hypothesis, song functions in resource defence and territorial disputes between individuals or groups, with these agonistic interactions driving the evolution and maintenance of female song. Territory defence represents one of the best-supported functions of female song across species, with females in several species using song similarly to males for solo, paired or group defence of a resource (reviewed in ref. 6), and has been previously shown to be

strongly associated with female song incidence[24]. Indeed, we found that strong territoriality and year-round territoriality were both significant predictors of female song presence. However, our phylogenetic path analyses showed that the association between cooperative breeding and female song persists even when controlling for territoriality, and our phylogenetic generalized linear models demonstrated that the bidirectional effects linking female song and cooperative breeding are strongest in weakly territorial species. Further, if social competition for territories or other resources was the primary force driving female song evolution, we would expect female song to be similarly associated with other high-density social systems that presumably intensify resource competition. However, we found that cooperative breeding was a stronger predictor of female song than colonial living and familial living for both methods of territory classification and that larger group sizes actually predicted an absence of female song. Together, these patterns suggest that, while social competition may drive the presence of female song in some contexts, it does not account for the specific association with cooperative breeding, especially in species with weak or absent territorial behaviour.

A third hypothesis is that female song could serve a role in maintaining social cohesion in cooperative breeding groups. Song produced in the context of established breeding pairs is already well documented as a factor in reinforcing pair bonds; however, evidence suggests that song can also mediate affiliative social interactions beyond the breeding pair, potentially by co-opting the reward systems that support pair-bond maintenance. For instance, European starlings' 'gregarious song', which occurs outside the breeding season, is positively correlated with social flocking behaviour and is thought to be intrinsically rewarding[35,36]. Male zebra finch 'undirected song' (that is, not directed towards a female) occurs year-round and is hypothesized to have prosocial functions, such as breeding synchronization, promoting social cohesion and individual recognition[37]. Female song shares some patterns with these examples (reviewed in ref. 4): it often occurs both in and out of the breeding season[38] and has been linked to breeding-related contexts, including coordination of breeding activities[4,39] and interactions with offspring after fledging[40], and non-breeding-related functions, including helper retention[41], individual or group-member recognition[41,42], neighbour recognition[43] and social cohesion[44]. These affiliative functions suggest that song could strengthen cooperative interactions through group cohesion, bonding and coordination in species where cooperation and altruism are key life history strategies. While social living may often lead to more frequent competition among females for mates or other resources, the maintenance of the social group—and the fitness benefits thereof—should

**Table 2 | Phylogenetic generalized linear model comparisons of cooperative breeding versus alternative social predictors of female song**

| Social variable | Number of species | Results from the model using cooperative breeding as the predictor (female song=cooperative breeding+territoriality+body mass) | | | | Results from model using the social variable as the predictor (female song=(social variable)+territoriality+body mass) | | | |
|---|---|---|---|---|---|---|---|---|---|
| | | Cooperative breeding model AIC | Cooperative breeding OR | Cooperative breeding OR 95% CI | Cooperative breeding P | Social variable model AIC | Social variable OR | Social variable OR 95% CI | Social variable P |
| Caretakers (normalized) | 242 | 251.8 | 4.65 | 1.57–15.29 | **0.008** | 241.1* | 8.33[a] | 2.16–31.09 | **0.032** |
| Familial living | 407 | 445.8 | 1.91[a] | 0.93–4.03 | 0.072 | 445.9 | 1.35 | 0.79–2.47 | 0.284 |
| Colonial living | 242 | 251.8* | 4.60[a] | 1.34–13.98 | **0.008** | 257.5 | 1.47 | 0.64–3.38 | 0.356 |
| Grouping (normalized) | 231 | 238.6 | 8.37[a] | 2.34–38.60 | **<0.002** | 240.1 | 0.68 | 0.48–0.93 | **0.02** |
| Social bond duration (normalized) | 242 | 251.8 | 4.57[a] | 1.28–14.71 | **0.016** | 252.0 | 1.54 | 1.18–2.09 | **<0.002** |

We compared cooperative breeding against each social variable using identical model structures with additive predictors (female song=(social variable)+territoriality (weak versus strong)+body mass). Sample sizes represent species with complete data for both cooperative breeding and the social variable being compared. All ordinal or continuous social variables were standardized (scaled to have an s.d. of 1 and mean of 0). The OR for each social variable predictor is reported as the mean and 95% CI across 500 bootstrap iterations. The P value was determined empirically based on the fraction of bootstrap iterations where the predictor's OR fell above or below 1.0; significant P values (P<0.05) are shown in bold. AIC values that are more than 2 ΔAIC lower than the alternative model are marked with a single asterisk. [a]The higher OR of the two models has been compared.

select for behaviours that minimize sexual conflict, particularly in systems with both high relatedness and frequent cooperation. While social bond duration and the number of caretakers also showed positive associations with female song, most other social variables showed weak or negative associations. This pattern, where cooperative caretaking and longer social bonds promote female song but other forms of group living do not, supports the hypothesis that female song may serve prosocial functions specifically in stable, cooperative social systems. Further, species where female song already mediates group coordination or offspring communication may be preadapted for transitions to cooperative breeding, as the vocal behaviours necessary for managing complex multi-caregiver systems would already be in place.

The sex-specific patterns in our data provide additional support for distinct evolutionary pressures on male and female song in cooperative species. Male song repertoire size evolved more slowly not only in cooperative breeding lineages but also in familial lineages, regardless of whether cooperative breeding was present. This pattern suggests that familial living and cooperative breeding can each stabilize male song elaboration, while only kin-cooperative breeding promotes or stabilizes female song. These contrasting results may hint at sex-specific functions and selection pressures underlying male and female song evolution. For example, our results could suggest that female song, but not male song, has group-cohesive functions because female song is most promoted in the system in which cohesion should have the greatest fitness benefit, that is, kin-cooperative systems[33]. Our results might also suggest that male song, more so than female song, is important for kin recognition and provides a mechanism for inbreeding avoidance, in which case male song repertoire size might be conserved in systems in which extended kin contact leads to a higher risk of inbreeding, that is, both kin-cooperative breeding and familial living[45].

Our findings regarding song repertoire evolution alongside other social factors may provide additional context. In addition to non-cooperative, non-familial systems, song repertoire size evolves more rapidly in pair-based or large-group social systems, lineages with shorter social bonds and colonial living systems, in addition to polygynous mating systems (identified in ref. 32). Altogether, these results suggest a mixed-selection-pressure paradigm in which male repertoire sizes evolve faster in environments where there is greater competition for mates because of increased social density and less social cooperation and are conserved when there is reduced competition among males for mates, shared effort among groupmates in defending territory or pressure against becoming too elaborate and disrupting cohesion within a stable group.

Exposure to challenging or variable environmental conditions may provide another explanation for the link between cooperative breeding and male song repertoire stability because complex songs are hypothesized to be disproportionately affected by stressful environmental conditions during development[46], and such conditions are also theorized to contribute to the emergence of cooperative breeding[47] or defensive behaviours[48]. The latter theory is bolstered by findings suggesting that cooperative breeding is more common in bird species found in environments with highly variable climates, such as sub-Saharan Africa, Australia and the tropics[49–51]. Several studies have also linked cooperative breeding with life history traits indicative of individual robustness, for instance, linking cooperative breeding with longer post-fledgling feeding periods and longer lifespans[52,53], both of which are expected to be linked with overall less stressful development. These trends could have implications for song evolution because developmental conditions can have important effects on young birds; a stressful upbringing can lead to impairments in both song learning and the development of the neural song system, while extended parental provisioning promotes the evolution of larger brain sizes across species[25,54]. Thus, extended parental care and cooperative breeding could help maintain song-learning capacity in harsh or highly variable environments, potentially stabilizing the song over time. Indeed, consistent with this hypothesis, we found slower song repertoire evolution in both cooperative breeding lineages and lineages with multi-year social bonds.

In future work, our findings could be clarified by recategorizing species with female song based on the level of sexual dimorphism in song features or production or specific contexts of female song, such as duetting. In the longer term, it would be highly informative to have species-level characterizations of female song in addition to male songs. Finally, it has been suggested that cooperative breeding should be considered a three-state trait, distinguishing species that exhibit occasional cooperative breeding from those in which it is either common or absent[55,56]; separating the cooperative breeding trait as such might reveal more complex trends. Additionally, expanding datasets for other social variables beyond cooperative breeding would enable more comprehensive analyses of how different forms of sociality influence vocal evolution.

In conclusion, we have found robust evidence for context-dependent associations between cooperative breeding and singing behaviour across the songbird phylogeny, with territorial systems mediating when and how these traits coevolve. While cooperative breeding and female song co-occur at high rates in strongly

territorial systems, their mutual reinforcement is especially pronounced in weakly territorial systems. Our findings also suggest that the context-dependent relationship between cooperative breeding and female song is best captured by territorial classifications that reflect how vigorously species defend territories rather than simply the duration that the territory is held. The difference in the relative effect of cooperative breeding in different territorial systems suggests that female song evolution follows multiple pathways: territorial defence in competitive environments and social coordination in cooperative contexts. While male song repertoire size evolution is constrained by several forms of stable social organization, female song shows robust associations with cooperative breeding that persist even when controlling for territoriality, body size or other life history factors. This relationship, combined with its context-dependent nature, suggests that female song can serve important functions in facilitating the coordination required in cooperative breeding systems. Our findings underscore the idea that song can have social functions beyond its canonical roles in mate choice and territory defence and reveal how cooperative living can shape communication systems in fundamentally different ways depending on the ecological and social context.

## Methods

All analyses were conducted in R v.4.3.1. Phylogenetic comparative methods used ape v.5.8, phytools v.2.3.0, phylolm v.2.6.5 and phylopath v.1.3.0. Data wrangling and visualization used the tidyverse v.2.0.0 (including dplyr v.1.1.4, tidyr v.1.3.1, stringr v.1.5.1 and ggplot2 v.3.5.1) and the associated plotting packages cowplot v.1.1.3, ggpubr v.0.6.0, ggrepel v.0.9.6, patchwork v.1.3.0, grid v.4.3.1 and gridExtra v.2.3. A full list of packages is provided in the publicly available codebase[57] (see Code Availability).

### Compiling the database

**Cooperative breeding.** We found previously published data on cooperative breeding using the exact search criteria "cooperative breeding in birds" and "songbird cooperative breeding" on Google Scholar and Web of Science, yielding 11 sources with previously compiled species information on cooperative breeding[10,16,33,49–52,55,58–60]. We considered cooperative breeding behaviour to be alloparenting during incubation, brooding and nest provisioning in 10% or more of nests in a study, if quantitative data were available, according to the classification methods of Cockburn[10]. We did not consider nest building, territory defence or fledgling provisioning by extrapair individuals to be sufficient for cooperative breeding classification. When a source's data were qualitative, species described unambiguously as 'cooperative', 'breed cooperatively', 'often cooperative', 'facultatively cooperative breeding' or as commonly having 'helpers' at the nest were considered to be 'cooperative', and species described with '(cooperative breeding) does not occur', 'none' and 'extremely rare' were considered to be 'non-cooperative'. We did not consider 'not observed', 'not known to occur' or 'never reported' to be sufficient evidence for the absence of cooperative breeding. If a source described a species as having 'occasional', 'possible', 'suspected', 'inferred', 'sometimes' or 'infrequent' cooperative breeding, we did not assign a binary classification to the species from that source[55]. Cooperative breeding classifications were assigned to each species using a binary system based on the aggregation of classifications from unique sources describing the species as cooperative or non-cooperative; when one source cited another source included in our dataset, we did not consider the citing source to be unique. Species for which source classifications were not unanimous were, when possible, classified based on the description in *Birds of the World*, otherwise they were omitted from our primary analyses (HighConfidence_Coop). To ensure that our results were robust, we repeated any analyses that produced significant results using alternative classification methods

based on whether most unique sources classified a species as cooperative versus non-cooperative. For those species that were 'tied' with equal numbers of sources classifying them as non-cooperative and cooperative, we ran secondary analyses with these species (1) classified as non-cooperative (MeanCoopTie2Noncoop), (2) classified as cooperative (MeanCoopTie2Coop) and (3) omitted from the dataset (MeanCoopOmitTies). We also ran analyses with (4) any species with at least one source classification of cooperative breeding designated as cooperative (AnyCoopEqualsCoop) and (5) any species with at least one source classification of non-cooperative breeding designated non-cooperative (AnyNoncoopEqualsNoncoop) (Supplementary Table 1). We also repeated analyses using the subset of classifications from several individual sources (Supplementary Dataset 2). We also tested the effect of recategorizing cooperative breeding based on the 'inferred' parental care mode, rather than the higher-confidence 'known' classifications from ref. 10.

**Other life history metrics.** We obtained species classifications for several metrics of sociality from ref. 25. We binarized several categorical metrics from that study related to group size, duration of social bonds, colonial living and number of caretakers (Supplementary Table 1). We also included binary classifications of species as socially monogamous or socially polygynous from ref. 32, which classified social polygyny primarily based on having an observed rate of polygyny above 10% in studied breeding pairs[32] and as having familial or non-familial living from ref. 16, which classified familial living as species where offspring remain with parents for at least 50 days after nutritional independence. We also included multistate traits for social group size from ref. 25 and the interaction of familial living and cooperative breeding from ref. 16. In our phylopath and phyloglm models, we incorporated log(body mass) from ref. 61.

**Territory.** We obtained territory data from ref. 21, which categorizes territorial behaviour into three states: non-territorial (level 1), seasonal or weak territoriality (level 2) and year-round territorial (level 3). However, the second level of this categorization groups species with strong seasonal territoriality together with species that are weakly territorial, which might lump species with very different territorial strategies while also separating species that exhibit strong territorial defence in temperate regions from those with year-round territory defence primarily in tropical regions. To better understand the relationship between our focal traits and the intensity of territoriality, we refined this classification into 'weak' versus 'strong' territorial defence categories. Non-territorial species (territory level 1) were assigned to the 'weak' category and year-round territorial species (territory level 3) to 'strong'; species with seasonal or weak territoriality (territory level 2) were individually recategorized by incorporating additional information from published sources. For detailed classification criteria, see the Supplementary Methods.

**Female song.** We combined and updated published datasets on the presence of female song in songbird species in natural conditions[2,62–64] (Supplementary Dataset 3), which we located by performing searches on Google Scholar and Web of Science for "female song passerine/passeriformes" and "female song songbird". We considered a species to have female song if it was classified as such in one of the published female song databases or if a source unambiguously said that females sing. Whenever sources disagreed on female song classification, we first attempted to resolve these discrepancies by cross-referencing with the species entry in *Birds of the World*[65]. If one of the sources for female song in a species with source discrepancies included Garamszegi et al.[63], we also cross-referenced these classifications with their original source in the encyclopaedia *The Complete Birds of the Western Palearctic*[66]. We also used data from Mikula et al.[67], but since the species classifications in that source created binary variables for female solo song, absence of

female song and duetting (a category that included chorusing species, which may include species in which only males sing), we only used species classifications that indicated that females produced solo song or did not sing. We further checked *Birds of the World* if a species was very well studied (that is, had at least one publication noting species song features and at least one publication regarding breeding behaviour in the species) but did not appear in any female song database. Species in which females had only ever been documented singing after treatment with estradiol or testosterone were omitted from our dataset, even if another source classified them as having female song present. Documentation of subsong in females in a species was not considered sufficient for a classification of female song being present. We were ultimately able to assign female song classifications for 1,094 species. While the diversity in female singing behaviours, song complexity and their difference from male songs are crucial areas for future research, for the purpose of maintaining sufficient sample sizes with the data available, we opted not to assign female song classifications along more granular categories related to song complexity or song difference from males.

**Song features.** We obtained species data on syllable repertoire size (average number of syllables a bird can produce), syllables per song (average number of unique syllables a bird sings per song), song repertoire size (the number of individual songs a bird knows), song duration (the length of a song) and inter-song interval (the time between songs) from Snyder & Creanza[32] and added several more species-level data points to the database (Supplementary Dataset 1). The primary sources of the song-feature data were generally field studies of presumably male birds; when song-feature data were specified according to sex in the literature, we used only the male song measurements. Whenever we obtained data for a species whose name was not present in the BirdTree (https://birdtree.org/) phylogeny, we searched Avibase for alternative taxonomic classifications[68]. If an alternative species name was present in the BirdTree phylogeny (for example, the genus or species name has been updated), we recorded the species data in our database under the BirdTree-matching name.

**Dimorphism.** We quantified both sexual dimorphism (body size) and sexual dichromatism (plumage coloration). For plumage dichromatism, we used species-specific and sex-specific scores from ref. 60, calculating the absolute difference between male and female plumage scores and applying a log transformation with an offset of 0.1 to handle zero values. We obtained a plumage dichromatism value for all of our 1,041 species with both cooperative breeding and female song data. For size dimorphism, we used morphometric data from the AVONET database[61], focusing on wing length as a proxy for body size. We log-transformed raw wing measurements, calculated sex-specific means for each species and quantified dimorphism as the percentage absolute difference between male and female log-transformed wing lengths relative to female values. We obtained wing dimorphism values for 987 of our 1,041 species with cooperative breeding and female song data.

### Generating a consensus phylogeny

We downloaded 1,000 trees randomly sampled from a pseudo-posterior distribution of trees using the Hackett backbone from BirdTree (https://birdtree.org/)[26,69]. We subsetted these trees to include only the 5,966 passerine species (including Oscines and Suboscines). We then built a consensus tree with branch lengths using consensus.edges() phytools v.2.3.0, with method = "mean.edge" and the default setting if.absent = "zero", which treats missing edges as zero in the mean edge calculation, but see 'Accounting for phylogenetic uncertainty' below for other approaches. For all subsequent analyses, we removed the Suboscine clade and only Oscine species were included in the reference phylogenies.

### Estimating rates of transition between states of categorical traits

We conducted ancestral character estimations with ace() ape v.5.8, using the maximum likelihood method to estimate the evolutionary history of discrete traits (for example, cooperative breeding, sociality metrics, female song presence) across all Oscine species for which we had a classification for a given trait. Using a likelihood ratio test, we compared the fits of the ancestral character estimation model that assumed 'equal rates' of transition between binary states and the model that assumed 'all rates different' (Supplementary Table 19). For traits with three or more states, we also tested the model that assumed 'symmetric' rates. In subsequent analyses, we used the all rates different model whenever it was significantly better than the equal rate model (and symmetric model, when applicable). Whenever we generated models of trait evolution on subsets of species, to avoid any potential bias from recalculating the rates for each species subset, we used the transition rates between states obtained from the full set of Oscine species that had a state classification for that trait.

### Song feature evolution in the context of cooperative breeding and other categorical traits

To determine whether there was a difference in the values of various song characteristics between groups of species that exhibit certain social traits, for example, cooperative and non-cooperative breeding species, we performed a set of analyses using phylanova() phytools v.2.3.0[70] to compare natural log-transformed song characteristics (obtained from Snyder & Creanza[32]) between the sets of species in the different social trait categories, with 50,000 simulations per song feature + social trait comparison[32].

We then tested whether any song characteristics evolved at different rates when in a cooperative breeding state versus a non-cooperative breeding state using the Brownie algorithm[71]. We generated 500 simulated stochastic character maps (simmaps), which simulate ancestral gains and losses of cooperative breeding given the transition rates calculated above, using make.simmap() phytools v.2.3.0, and used them with brownie.lite() phytools v.2.3.0. We performed follow-up Brownie analyses using each of the binarized sociality traits and song repertoire size. We further performed Brownie analyses on song repertoire size evolution and the multistate traits of group size ('grouping' from ref. 25), the intersection of familial living and cooperative breeding traits from ref. 16 and the intersection of our cooperative breeding and territoriality strength classifications.

Species for which there were multiple values for a certain song feature in the literature had their minimum and maximum published values recorded in the feature dataset. To account for potential variations in song feature measurement across species, researchers, locations and decades of study, for the feature that showed significant dependent evolution (cooperative breeding with song repertoire size), we repeated the Brownie analysis with only the minimum values, with only the maximum values and with random sampling from the minimum, median and maximum values for all species, where we resampled the song repertoire values 500 times, generating two simmaps per resampled vector to conduct the analyses of evolutionary rates.

### Analysing evolutionary co-occurrence of cooperative breeding, female song, territoriality and other traits

To test whether pairs of discrete traits, including cooperative breeding, other sociality classifications (Supplementary Table 1), and female song presence, which co-occurred in the songbird lineage more often than expected by chance, we used the methods adapted from ref. 27. This process involved generating simmaps using make.simmap() for the two discrete characters in question (for example, cooperative breeding and female song) and setting the rates of transition between states ($Q$) to those estimated using ace(). Then, for each pair of simmaps, we calculated the proportions of the branches of the phylogeny spent

in each of the four-state combinations arising from combinations of the two binary traits (or more than four when testing one of the multistate traits versus female song) using phytools::map.overlap(). We then calculated the statistic $D$ (equations given in ref. [27]) to obtain an overall measure of the disagreement between observed and expected proportions of each combination of characters. Per ref. [27], we repeated this process to generate simulations of the two discrete traits evolving independently from one another by randomly shuffling the tip states at the same proportion as the actual species classification data, then generating each simmap with make.simmap(). The empirical $P$ value was calculated as the fraction of randomized data simmap pairs with a $D$ statistic larger than the median $D$ statistic of the simmap pairs generated from the real data. Unless otherwise noted, we performed 500 real and 500 simulations with randomized data per trait comparison. We also note that the methods of ref. [27] can be misleading when one or more of the studied traits are clustered in one small section of the tree or have one major origin in evolutionary history[72]; however, when we visualized these traits on a large phylogeny, we observed many gains and losses of both female song and cooperative breeding throughout the tree (Fig. [2]).

## Testing evolutionary associations and interactions using simulated state transitions

We used the simmaps generated using the real tip states to estimate whether the state of one trait influenced the likelihood that certain transitions occurred in the other trait. For each pair of trait simmaps, denoted here as trait A and trait B, we identified every state transition in each direction (trait $A_{0\to1}$ and trait $A_{1\to0}$) on the trait A tree, then identified which state was present at the same point on the trait B tree and vice versa. This gave us eight observed transition counts, 'trait $A_{0\to1}$ in trait $B_0$', 'trait $A_{0\to1}$ in trait $B_1$', 'trait $A_{1\to0}$ in trait $B_0$', 'trait $A_{1\to0}$ in trait $B_1$', etc. We calculated the expected number of transition counts for each simmap pair by multiplying the total number of trait A transitions in each direction by the fraction of the tree spent in each state of trait B and vice versa. We log-transformed these state switch counts across the 500 simulations and fitted a two-way linear model with count type (observed versus expected) and state transition type (eight possible transition types per pair of binary traits) as fixed factors, including their interaction. An ANOVA on this model tested whether the difference between observed and expected counts varied among transition categories. When the interaction term was significant, we conducted Tukey-adjusted pairwise comparisons of estimated marginal means within each transition type to identify which specific transitions differed significantly from the expected counts. We applied this transition-count framework to female song paired with each binary social variable (Supplementary Fig. 2). To assess the effect of territoriality on the interactions between cooperative breeding and female song, we also generated a third set of simmaps for our binary weak versus strong territoriality trait and quantified the transitions between female song and cooperative breeding states depending on the territoriality state at the corresponding point on the third simmap.

## Testing evolutionary associations and calculating effect sizes using phylogenetic generalized linear modelling

We also performed phylogenetic logistic regressions using phyloglm() phylolm v.2.6.5[73,74] to test for evolutionary associations between traits in several ways.

Primarily, we tested for associations between female song presence and cooperative breeding while controlling for territoriality (binary measures of territory strength and year-round territoriality) and body mass (log-transformed, centred and scaled) on our full dataset. We fitted multiple competing models with different interaction structures and used AIC model selection to identify the best-supported trait associations. For the best-fit model, we ran 500 bootstrap

iterations to quantify uncertainty in the effect sizes of the predictors; we report the OR, 95% OR CI and $P$ value for each predictor.

We integrated additional variables through two additional methods: (1) direct comparison of identical models that include either cooperative breeding or an associated social variable as the main predictor, alongside territoriality and body mass, to determine whether any social variables better predicted female song than cooperative breeding; and (2) forward stepwise model expansion to assess whether any life history variables improve or alter the best-fit models predicting female song and cooperative breeding. For each model discussed in this article, we performed 500 bootstrap iterations and we report the OR, 95% CI and $P$ value for the parameters in question.

First, to address whether other social variables might better predict female song than cooperative breeding, we performed direct model comparisons. For each social variable (colonial living, familial living, group size, social bond duration, number of caretakers), we compared models using that variable versus cooperative breeding (alongside territoriality and body mass) as predictors, applied to the subset of species that had data for both cooperative breeding and the social variable in question. Notably, as the number of species for which there are data available for the various social variables is much smaller than our full cooperative breeding female song dataset, the relative performance of cooperative breeding versus other social variables should be interpreted with caution because the reduced datasets may not capture the full phylogenetic or ecological diversity of our complete sample.

Second, we integrated additional life history variables through a forward stepwise selection process. Starting with the best model from each of the four main analyses (female song or cooperative breeding as the response variable; territory strength or year-round territoriality as the territoriality variable; best model determined as the lowest AIC), we iteratively tested the addition of candidate predictors: absolute latitude (range centroid, centred and scaled[61]), wing dimorphism (percentage absolute difference in log-transformed wing length between sexes, centred and scaled[61]), plumage dichromatism (log-transformed absolute difference in male–female plumage scores, centred and scaled[60]), migration (three-level, categorical: sedentary, partially migratory, migratory, treated as numeric[61]), familial living (binary[16]), social mating system (binary social monogamy versus polygyny[32]) and geographical region (binary: Holarctic versus Tropical[49]). At each iteration, we added each remaining candidate predictor to the current model, refitted both the base and the expanded models using only species with complete data for all variables and selected the predictor that yielded the greatest AIC improvement. We accepted predictors that improved model fit by $\Delta$AIC $\geq 2$ and continued iterations until no remaining predictor met this threshold (Supplementary Tables 21–24).

## Testing correlated evolution and causal relationships with phylogenetic path analysis

We used phylogenetic path analysis (R package 'phylopath'[75] 1.3.0) to test for correlated evolution among female song presence, cooperative breeding, territoriality and body size. This method compares alternative causal models while controlling for phylogenetic non-independence, allowing us to test whether these traits evolved independently or showed evolutionary interdependence. We specified competing models representing different hypothetical causal relationships (for example, cooperative breeding → female song versus female song → cooperative breeding versus independent evolution) and used CICc model selection to identify the best-supported evolutionary scenario. For each set of covariates, we calculated model-averaged standardized path coefficients across all models with a CICc value within two units of the best-supported model, with coefficients weighted according to relative model support. All models were tested both with body mass included as a covariate to control for known and hypothesized allometric effects and without body mass. (The full set of models is provided in Supplementary Fig. 1.)

## Assessment of data biases

To address potential systematic biases in data availability, we tested whether the species that had classifications for female song and cooperative breeding varied systematically across ecological, behavioural, morphological and geographical categories. In other words, we tested whether female song and cooperative breeding data were more likely to be collected in certain contexts (for example, from species with readily distinguishable males and females, from species that defend clear territories), reflecting biases in human research effort rather than avian biology. We used chi-squared tests to examine the associations between data availability and binary traits, including cooperative breeding status, coloniality[25], mating system (monogamous versus polygynous), territoriality classifications and geographical region (Holarctic versus Tropical, based on ref. 49, Supplementary Table 25). For sexual dichromatism, we calculated the absolute difference between male and female plumage scores from ref. 60 and tested for differences in dichromatism between species with and without female song data using Welch's $t$-tests. We approximated size dimorphism by averaging according to species all wing length measurements available for males and females in the AVONET database[61], log-transforming mean wing lengths according to species and sex, subtracting the female mean from the male mean, taking the absolute value and dividing by the female mean to get the percentage dimorphism per species. We also tested whether species with female song present versus absent differed in sexual dichromatism or size dimorphism because this could indicate detection bias where sexually dimorphic species are more likely to have confident female song classifications. Notably, among species with female song data, those classified as having female song absent had higher dimorphism than those with female song present (plumage: $t = 3.83$, $P < 0.001$; wing: $t = 4.18$, $P < 0.001$; Supplementary Table 20).

For those traits that did show significant biases in female song or cooperative breeding data availability, we performed repeated phylopath analyses, with models including cooperative breeding, female song, weak versus strong territoriality, and body mass (log-transformed) by downsampling the dataset to randomly remove species from groups that we found to be overrepresented. We downsampled the following groups in turn: Holarctic non-cooperative species, tropical cooperative breeding species, global cooperative breeding species and both wing dimorphism and plumage dichromatism (Supplementary Table 14). For the continuous variable dimorphism metrics, we performed downsampling corrections by randomly removing dimorphic species in proportion to their excess dimorphism above the mean of species without female song data, repeating phylogenetic path analyses across 500 iterations.

## Testing whether observed trends are driven by individual families

We repeated all analyses comparing song traits and cooperative breeding with significant results by performing a jackknife analysis according to family, that is, iteratively removing each family from the dataset and recalculating the results to ensure that no single family was driving the observed trends. Family classifications were obtained from AVONET[61].

## Accounting for phylogenetic uncertainty

We built a second consensus tree from the same sample of 1,000 trees from BirdTree (Hackett-Stage2, trees 3,001–4,000) using consensus. edges() with method "mean.edge" but ignoring missing edges in the edge length calculations (if.absent = "ignore"). To test whether this alternate method of calculating branch lengths meaningfully affected the results, we used this consensus tree to repeat tests that found significant or otherwise notable associations between song features and cooperative breeding and between female song and cooperative breeding. To further account for phylogenetic uncertainty, we re-ran these same analyses on 200 individual trees from BirdTree, with 20 simulations per tree. For these, we used the transition rates calculated using the default consensus tree to generate stochastic character maps on each individual sampled tree.

## Reporting summary

Further information on research design is available in the Nature Portfolio Reporting Summary linked to this article.

## Data availability

Data are publicly available at https://doi.org/10.6084/m9.figshare.26110339 (ref. 57).

## Code availability

Code is publicly available at https://doi.org/10.6084/m9.figshare.26110339 (ref. 57). Any future updates to the code will be made available to the public via GitHub repository (github.com/CreanzaLab/CooperativeBreedingEvolution).

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

## Acknowledgements

We thank members of the Creanza lab for helpful feedback and the field biologists who collected the behavioural data we compiled for the dataset. K.T.S. and N.C. were funded by grants from the National Science Foundation (IOS-2327982, BCS-1918824 and DUE-1926794). A.L.-P. was supported by the School for Science and Math at Vanderbilt. K.T.S., A.L.-P. and N.C. were supported by a Pilot Grant from the Evolutionary Studies Initiative at Vanderbilt.

## Author contributions

K.T.S. and N.C. conceived and designed the experiments. K.T.S., A.L-P. and N.C. contributed to the database and analysed the data. K.T.S. performed the experiments. K.T.S and N.C. produced the visualizations and wrote the paper.

## Competing interests

The authors declare no competing interests.

## Additional information

**Extended data** is available for this paper at https://doi.org/10.1038/s41559-026-02981-y.

**Correspondence and requests for materials** should be addressed to Nicole Creanza.

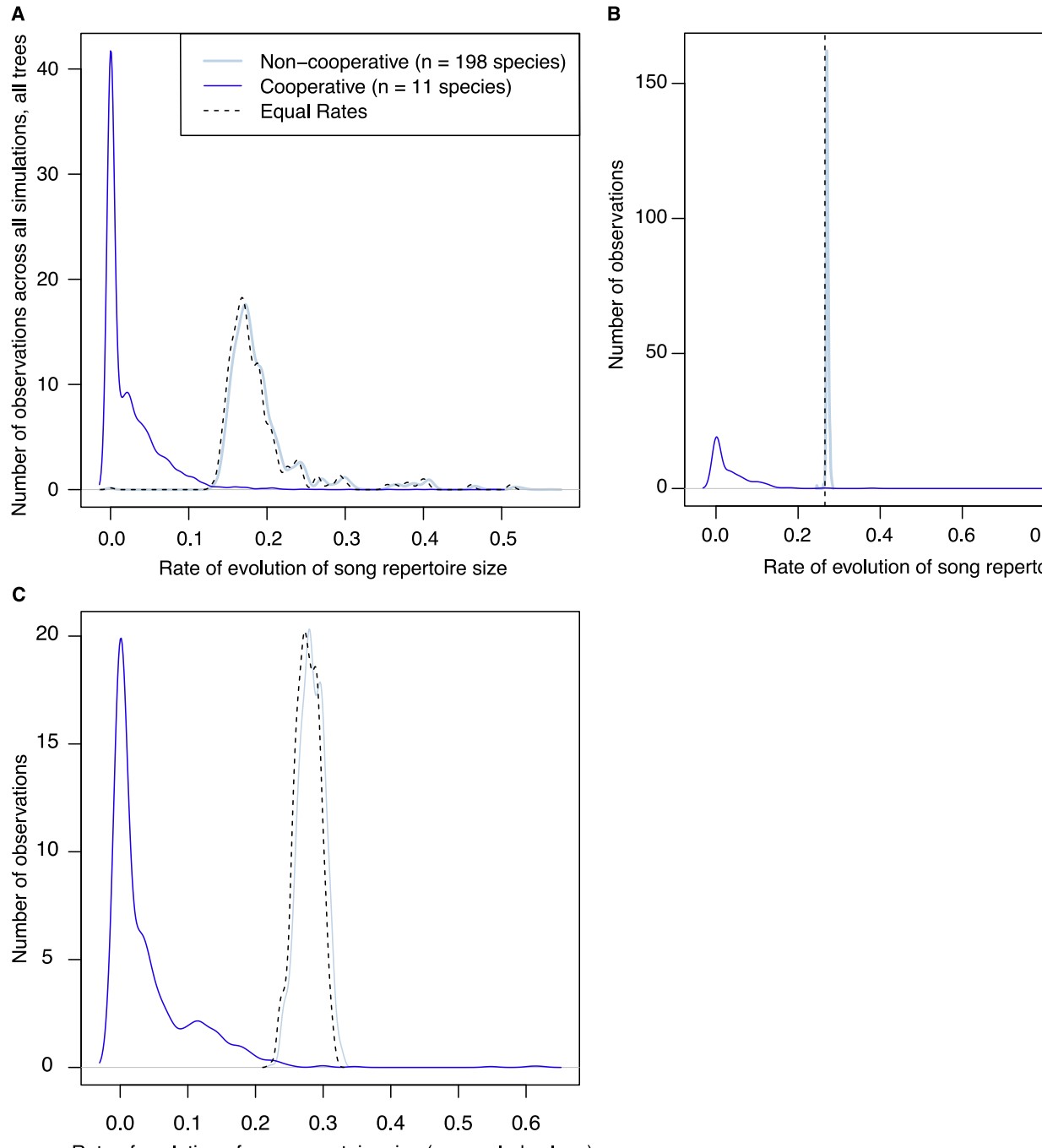

**Extended Data Fig. 1 | Rates of evolution of song repertoire size relative to cooperative breeding state using alternative methods to account for phylogenetic uncertainty.** We calculated these rates of evolution **A**) using 200 trees sampled from BirdTree.org (Hackett backbone, Stage 2), with 20 simulations per tree ($p = 0.032$); **B**) using an alternative consensus tree (calculated with absent edges ignored in the mean edge length calculations, $p = 0.030$); and **C**) using 500 iteratively resampled minimum/median/maximum song repertoire values, with 2 simulations per resampled data vector ($p = 0.0332$). n = 209 species. The legend in (**A**) applies to all panels. $p$-values are derived from likelihood ratio tests (one-sided, df=1) based on the median log-likelihoods for the more complex and more simple models across all simulations for each analysis. See Table S4.

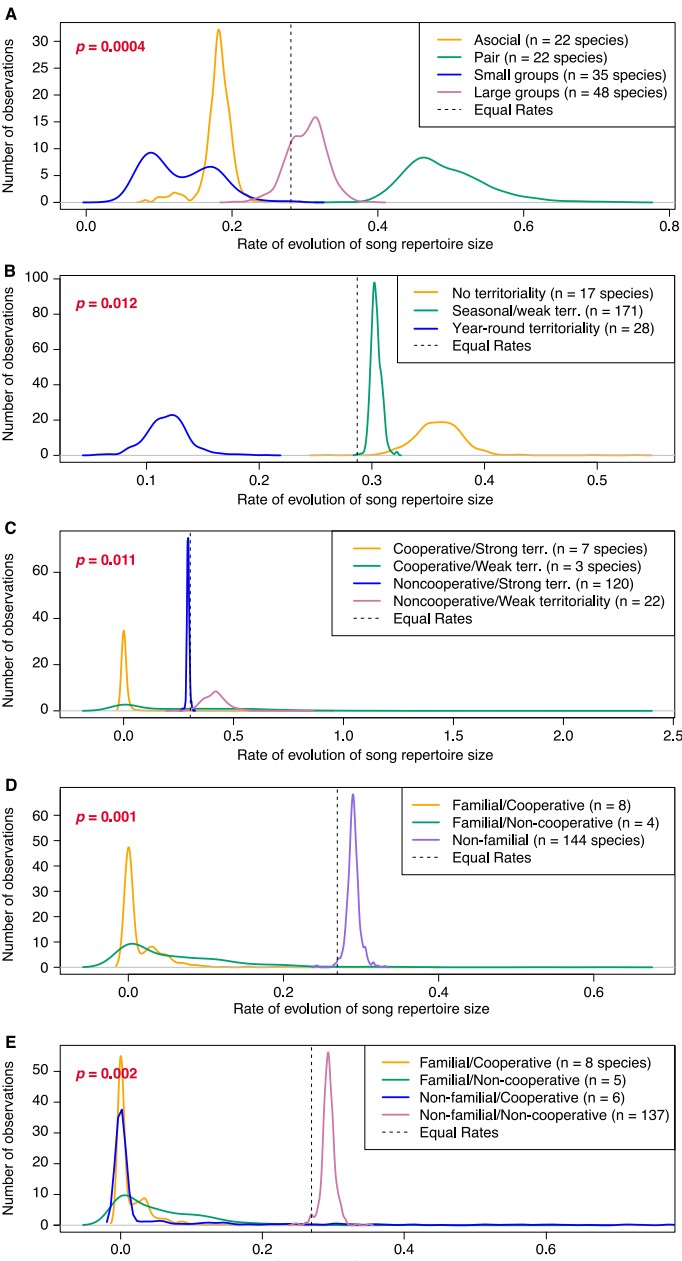

**Extended Data Fig. 2 | Results assessing whether the rate of evolution of song repertoire size differs based on the predicted ancestral state of various social factors.** These factors were (**A**) group size (from Griesser et al. 2023), (**B**) territoriality (from Tobias et al. 2016), and (**C**) the combination of cooperative breeding and weak vs strong territoriality, (**D**) a social system trait that could take three possible values, "Familial/Cooperative", "Familial/Non-cooperative", and "Non-familial" ("social system" from Griesser et al. 2017) or (**E**) a social system that could take four possible values, "Cooperative, Familial", "Cooperative,

Non-familial", "Non-cooperative, Familial", and "Non-cooperative, Non-familial" ("social system including nonkin-cooperative breeding" from Griesser et al. 2017). Each of the five analyses were performed for 500 simulations of the Brownie algorithm. *p*-values on each panel were derived from likelihood ratio tests (one-sided, df=1) based on the median log-likelihoods for the more complex and more simple models for each analysis. No adjustments were made for multiple comparisons. Sample sizes: (**A**) n = 127 species, (**B**) n = 216 species, (**C**) n = 152 species, (**D**) n = 156 species, (**E**) n = 156 species. See Table S6.

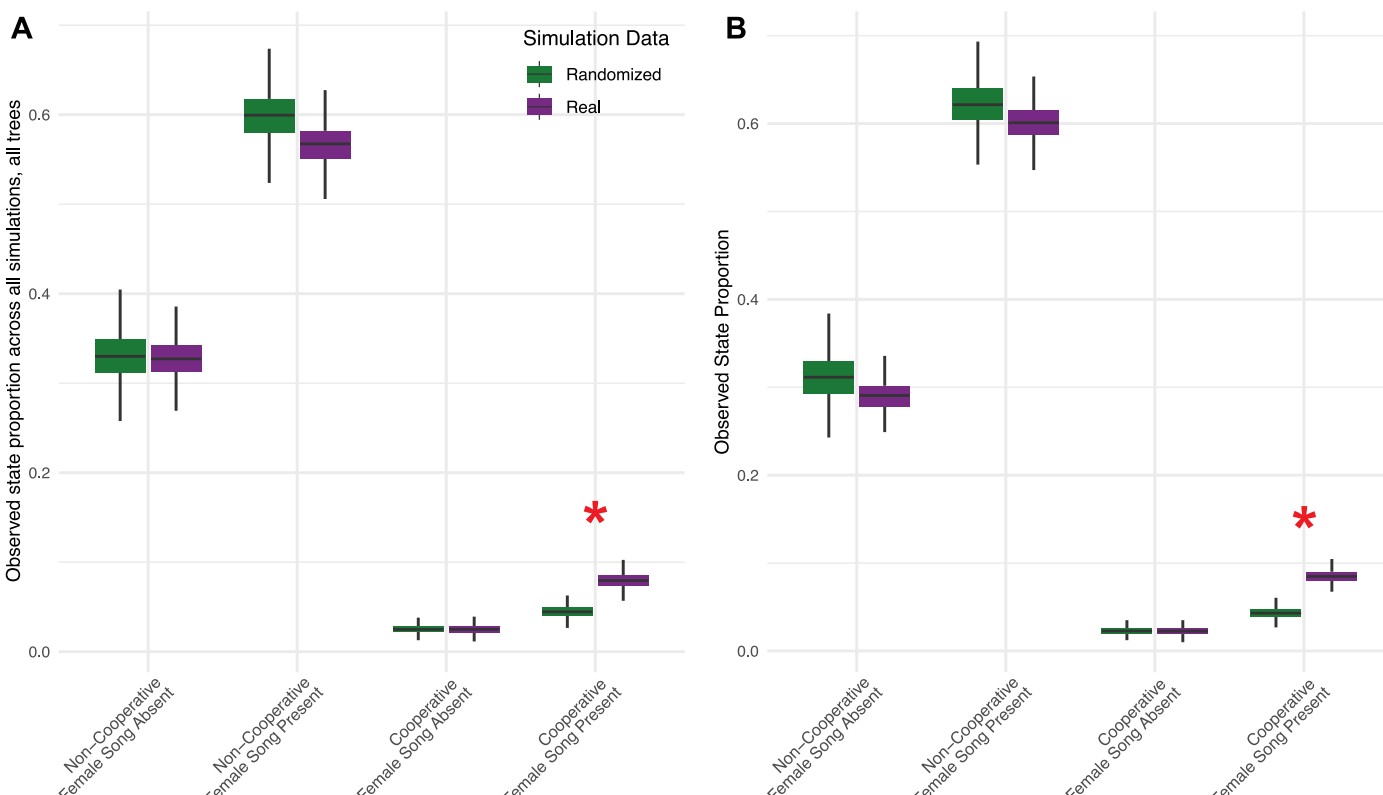

**Extended Data Fig. 3 | Overlap of stochastic character map simulations of female song and cooperative breeding, measured as the proportion of the tree occupied by each state combination. A)** Pooled results of analyses performed on 200 individual trees from BirdTree.org with 20 simulations per tree are shown and correspond with an overall empirical *p*-value of 0.00075. **B)** Results of analysis performed on the alternate consensus tree for 500 simulations with an empirical *p*-value of 0.008. Statistical significance was determined by simulation tests (two-sided, empirical p-values) comparing simulated overlaps based on observed data to simulations based on randomized (independent) data. A red asterisk indicates that at least 95% of the values from the randomized simulations were less than the median value of the simulations of ancestral history based on the true data. Sample size: n = 1041 species. Box plots show the median (center line), the 25th and 75th percentiles (box bounds), and whiskers extending to the smallest and largest values within 1.5x the interquartile range from the lower and upper quartiles; values outside this range (outliers), if present, are not displayed. See Table S8.

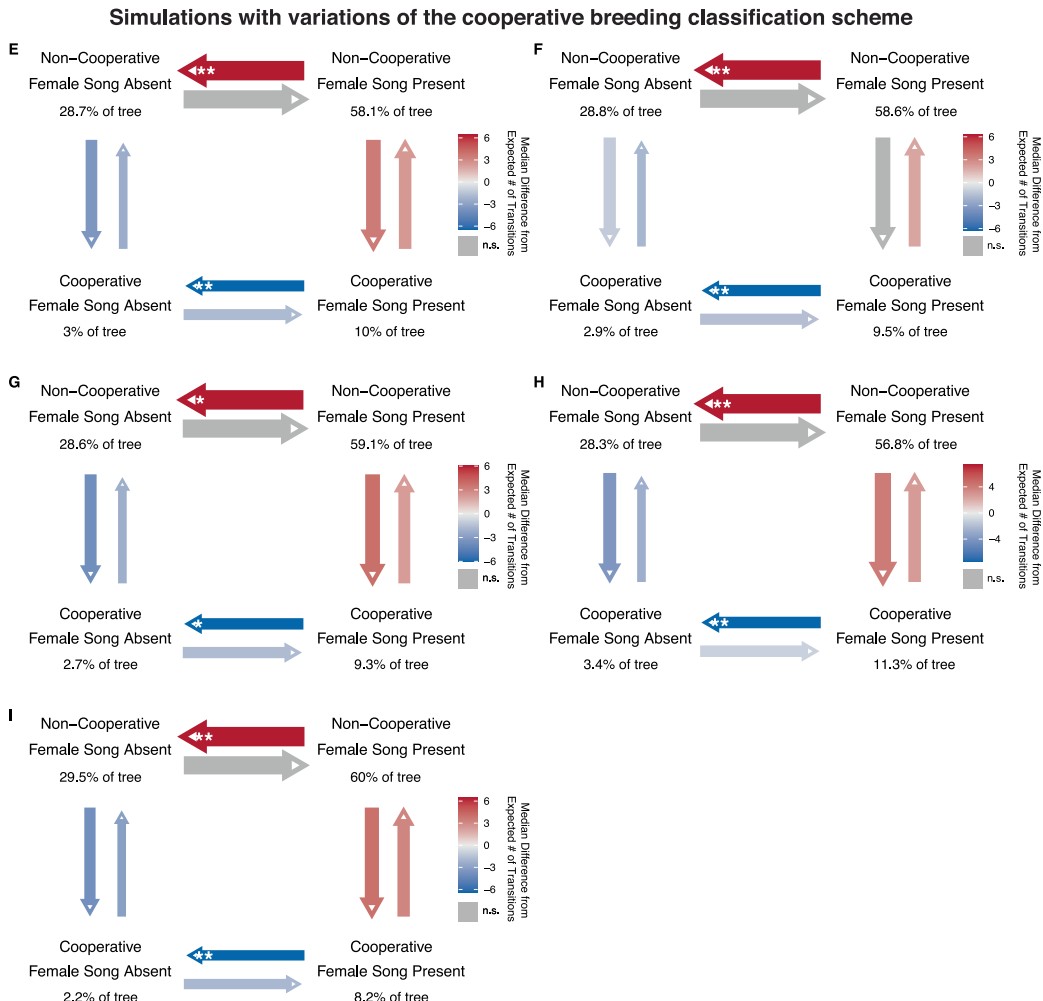

## Simulations with variations of the territoriality classification

Transition counts within non-/weakly/seasonally territorial lineages

Transition counts within year-round territorial lineages

## Simulations with variations of the phylogeny

## Simulations with variations of the cooperative breeding classification scheme

**Extended Data Fig. 4 | See next page for caption.**

**Extended Data Fig. 4 | Female song and cooperative breeding transition likelihoods based on the median difference from expected transition counts, calculated using alternate territory classification, alternative phylogenies to account for phylogenetic uncertainty, or alternative cooperative breeding classification methods.** Using the default cooperative breeding classification method: Panels **A** and **B** show the numbers of transitions between female song states and cooperative breeding states that occurred in lineages that were **A**) non-territorial or weakly/seasonally territorial and **B**) year-round territorial. **C**) Pooled results over 200 trees, 20 simulations per tree. **D**) Results from 500 simulations using the real trait distribution on the consensus phylogeny obtained from using the "mean.edge" method with absent edges ignored in the mean edge length calculations. Using the default consensus tree: **E**) MeanCoopTie2Noncoop, **F**) MeanCoopTie2Coop, **G**) MeanCoopOmitTies, **H**) AnyCoopEqualsCoop, and **I**) AnyNoncoopEqualsNoncoop. Arrow weights indicate the log-transformed median relative number of state switches that occurred on that topography. Deviations of observed from expected transition counts were assessed across stochastic character map simulations using the transition-count framework described in Methods: log-transformed observed vs. expected counts were analyzed with a two-way linear model (observed vs expected × transition), with significance assessed by ANOVA and Tukey-adjusted post-hoc comparisons within each transition. Grey arrows indicate transitions not significantly different from expectation; red/blue arrows indicate significantly more/fewer transitions than expected. Otherwise, color denotes the median difference from the expected number of transitions across simulations, with red indicating that the observed number of state switches was greater than expected, and blue indicating fewer state switches than expected. Asterisks indicate that the percent of simulations where the number of transitions is greater (red arrows) or less (blue arrows) than the calculated expected number of transitions for the simulation in question in >95% of simulations (**) or 90-95% of simulations (*). Sample sizes: **A**) n = 637 species, **B**) n = 404 species, **C**) n = 1041 species, **D**) n = 1041 species, **E**) n = 1075 species, **F**) n = 1075 species, **G**) n = 1066 species, **H**) n = 1075 species, **I**) n = 1075 species.

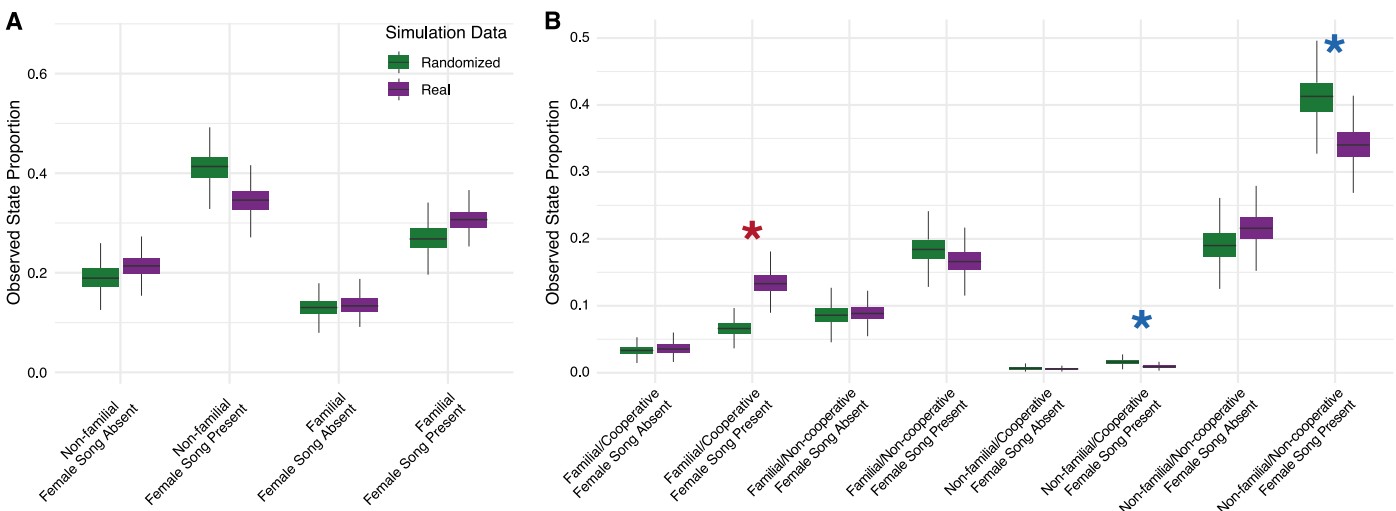

**Extended Data Fig. 5 | Overlap of stochastic character map simulations of female song and categorical social traits, measured as the proportion of the tree occupied by each state combination.** These social traits were **A**) familial living (based on data from[16] and female song (empirical *p*-value 0.092, 500 simulations, N = 468 species), and **B**) social system classification, encompassing both cooperative breeding and familial living[16], and female song (empirical *p*-value 0.0173, 1500 simulations, N = 468 species). Statistical significance was determined by simulation tests (two-sided, empirical *p*-values) comparing simulated overlaps based on observed data to simulations based on randomized (independent) data. Results of post-hoc analyses comparing real and randomized simulations are shown in panel B; a red asterisk indicates that at least 95% of the values from the randomized simulations were less than the median value of the simulations of ancestral history based on the true data, while a blue asterisk indicates that at least 95% of the randomized simulation values were higher than the median value of the simulations of ancestral history based on the true data (see Extended Data Table 1, Table S11). Box plots show the median (center line), the 25th and 75th percentiles (box bounds), and whiskers extending to the smallest and largest values within 1.5x the interquartile range from the lower and upper quartiles; values outside this range (outliers), if present, are not displayed.

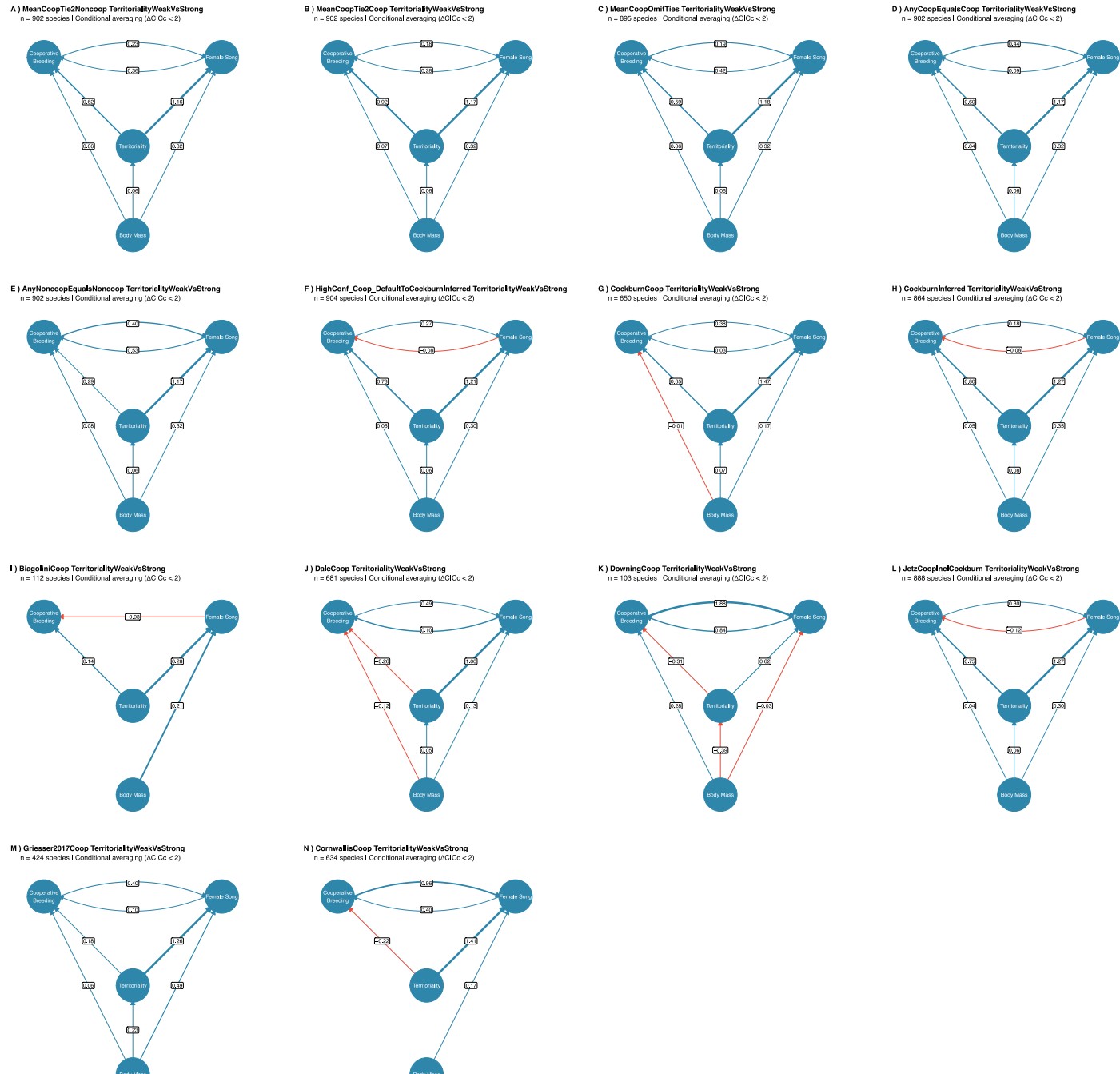

**Extended Data Fig. 6 | Robustness of phylogenetic path analysis results to different cooperative breeding classifications.** Directed acyclic graphs show the conditionally averaged path coefficients from phylogenetic path analyses of female song presence, cooperative breeding, territory strength, and body mass (log-transformed and normalized). Numbers on arrows are path coefficients, and arrow thickness scales with coefficient magnitude. Blue edges indicate coefficients >0, whereas red edges indicate coefficients <0. All coefficients shown are conditional averages across models with ΔCICc < 2 for the corresponding dataset, with sample sizes shown above each panel. Panels **A-F** use alternative cooperative breeding classification schemes based on agreement across sources. Panels **G-N** use cooperative breeding classifications derived from individual sources.

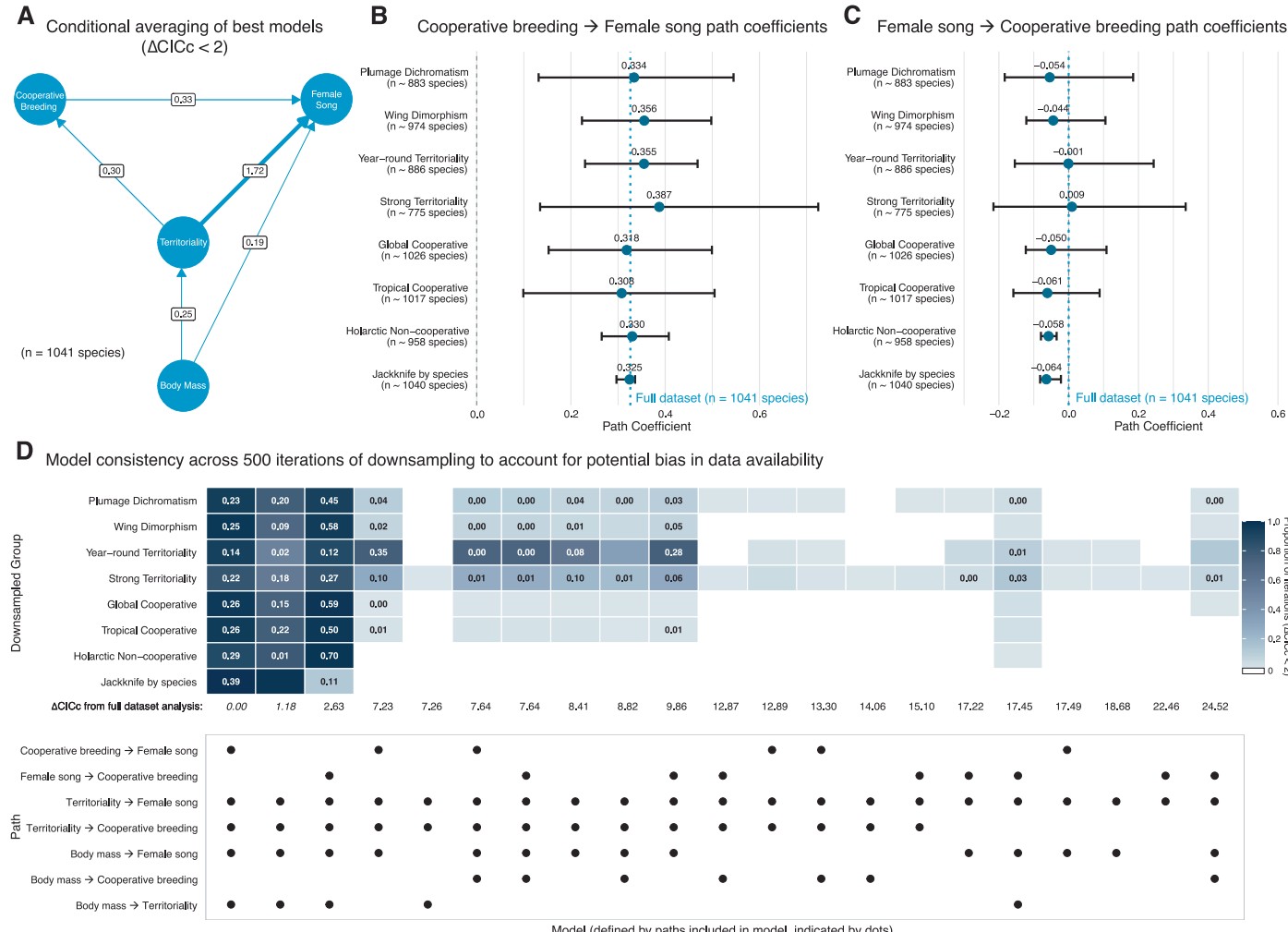

**Extended Data Fig. 7 | Robustness of the cooperative breeding-female song association to data availability biases using phylogenetic path analysis, with year-round territoriality and body mass as additional factors. (A)** Directed acyclic graph showing conditional-averaged path coefficients from phylogenetic path analysis of the relationships between female song presence, cooperative breeding, year-round territoriality, and body mass (log-scaled and normalized). This figure can be compared to the Fig. 4 main text, which shows phylogenetic path analyses with strength of territoriality included instead of year-round territoriality shown here. Arrow thickness indicates the magnitude of path coefficients, with all paths shown representing positive relationships. Only models within ΔCICc < 2 (2 models) were included in the averaging. **(B)** Forest plot showing the mean and 95% confidence interval for the cooperative breeding → female song path coefficient across 500 iterations for each bias correction type, compared to the value from the full dataset. **(C)** Forest plot

showing the mean and 95% confidence interval for the female song → cooperative breeding path coefficient across 500 iterations for each bias correction type, compared to the value from the full dataset. **(D)** Model consistency across bias corrections. Heatmap shows the proportion of iterations in which each phylogenetic path model was supported (had ΔCICc < 2) across 500 iterations for each bias correction type. Cells are labeled with the proportion of downsampled simulation iterations in which the model was the best-supported. The row underneath the heatmap contains the ΔCICc values for each model in the phylopath analysis performed using the full database, with the ΔCICc values of the two models that were statistically equivalent (ΔCICc < 2) in the full dataset in italics. The lower panel shows the models based on the presence or absence of each rate in the model. The models shown are all those that were in the top 20 models by # of iterations with ΔCICc < 2 for any of the bias correction types.

**Extended Data Table 1 | Results of overlapping simulations of ancestral character history for binary social variables and female song**

| Social Trait (Trait B) | Number of species | Empirical p-value | Degree of association between Trait A (female song absent versus female song present) and Trait B | | | |
|---|---|---|---|---|---|---|
| | | | Female song absent co-occurrence with Social Trait = 0 | Female song present co-occurrence with Social Trait = 0 | Female song absent co-occurrence with Social Trait = 1 | Female song present co-occurrence with Social Trait = 1 |
| Asocial (0) vs. social groups (1) | 255 | 0.438 | 0.432 | 0.36 | 0.548 | 0.608 |
| Asocial or pair (0) vs larger group (1) | 255 | 0.288 | 0.658 | 0.454 | 0.442 | 0.528 |
| Asocial, pair, or small group (0) vs. large group (1) | 255 | 0.316 | 0.392 | 0.464 | 0.772 | 0.494 |
| Social bonds lasting one season or less (0) vs longer than one breeding season (1) | 270 | 0.126 | 0.522 | 0.012 | 0.552 | 1 |
| Non-familial (0) vs. familial living (1) | 468 | 0.092 | 0.82 | 0.028 | 0.568 | 0.908 |
| Non-colonial (0) vs. colonial living (1) | 270 | 0.374 | 0.596 | 0.294 | 0.552 | 0.684 |
| Monogamy (0) vs. polygyny (1) | 323 | 0.248 | 0.128 | 0.716 | 0.758 | 0.62 |
| One or two caretakers (0) vs. more than two caretakers (1) | 270 | 0.156 | 0.586 | 0.214 | 0.398 | 0.968 |
| Weak or no territoriality (0) vs. strong territoriality | 919 | 0.002 | 0.992 | 0.010 | 0.260 | 0.706 |
| Weak, seasonal, or no territoriality (0) vs. year-round territoriality (1) | 1091 | <0.002 | 0.998 | 0.056 | 0 | 0.986 |
| Non-cooperative (0) vs. cooperative breeding (1) | 1041 | 0.002 | 0.274 | 0.194 | 0.504 | 1 |

For each pair of traits (female song + a social variable from the left column), we ran 500 ancestral character mapping simulations with the real data and 500 with shuffled data (i.e. with tip states in a randomized order) for comparison. We used methods adapted from[27] to estimate an empirical p-value (yellow indicates significance at p<0.05). Across the Trait A-Trait B stochastic character map pairs based on the real data, we found the median proportion of the phylogeny that each combination of traits occupied, then approximated the degree of association between trait states by comparing the proportion of the tree occupying a given trait combination in the real versus randomized simulations. Cells highlighted in blue indicate that that trait combination was less than the median of real simulations in <5% of randomized simulations (and thus the trait combination is less common than expected if traits were independent). Cells highlighted in red indicate that >95% of randomized simulations were less than the median of the real simulations (thus the trait combination is more common than expected if traits were independent).

# Reporting Summary

## Statistics

For all statistical analyses, confirm that the following items are present in the figure legend, table legend, main text, or Methods section.

| n/a | Confirmed | |
|---|---|---|
| ☐ | ☒ | The exact sample size (*n*) for each experimental group/condition, given as a discrete number and unit of measurement |
| ☐ | ☒ | A statement on whether measurements were taken from distinct samples or whether the same sample was measured repeatedly |
| ☐ | ☒ | The statistical test(s) used AND whether they are one- or two-sided *Only common tests should be described solely by name; describe more complex techniques in the Methods section.* |
| ☐ | ☒ | A description of all covariates tested |
| ☐ | ☒ | A description of any assumptions or corrections, such as tests of normality and adjustment for multiple comparisons |
| ☐ | ☒ | A full description of the statistical parameters including central tendency (e.g. means) or other basic estimates (e.g. regression coefficient) AND variation (e.g. standard deviation) or associated estimates of uncertainty (e.g. confidence intervals) |
| ☐ | ☒ | For null hypothesis testing, the test statistic (e.g. *F*, *t*, *r*) with confidence intervals, effect sizes, degrees of freedom and *P* value noted *Give P values as exact values whenever suitable.* |
| ☒ | ☐ | For Bayesian analysis, information on the choice of priors and Markov chain Monte Carlo settings |
| ☐ | ☒ | For hierarchical and complex designs, identification of the appropriate level for tests and full reporting of outcomes |
| ☒ | ☐ | Estimates of effect sizes (e.g. Cohen's *d*, Pearson's *r*), indicating how they were calculated |

*Our web collection on statistics for biologists contains articles on many of the points above.*

## Software and code

Policy information about availability of computer code

| Data collection | Code and data are available at https://doi.org/10.6084/m9.figshare.26110339 . Datasets have also been uploaded as supplemental materials. Data collection procedures are described in the Methods as well. |
|---|---|
| Data analysis | Code and data are available at https://doi.org/10.6084/m9.figshare.26110339 |

For manuscripts utilizing custom algorithms or software that are central to the research but not yet described in published literature, software must be made available to editors and reviewers. We strongly encourage code deposition in a community repository (e.g. GitHub). See the Nature Portfolio guidelines for submitting code & software for further information.

## Data

Policy information about availability of data

All manuscripts must include a data availability statement. This statement should provide the following information, where applicable:
- Accession codes, unique identifiers, or web links for publicly available datasets
- A description of any restrictions on data availability
- For clinical datasets or third party data, please ensure that the statement adheres to our policy

| Code and data are available at https://doi.org/10.6084/m9.figshare.26110339 |
|---|

# Research involving human participants, their data, or biological material

Policy information about studies with [human participants or human data](). See also policy information about [sex, gender (identity/presentation), and sexual orientation]() and [race, ethnicity and racism]().

| | |
|---|---|
| Reporting on sex and gender | n/a |
| Reporting on race, ethnicity, or other socially relevant groupings | n/a |
| Population characteristics | n/a |
| Recruitment | n/a |
| Ethics oversight | n/a |

Note that full information on the approval of the study protocol must also be provided in the manuscript.

# Field-specific reporting

Please select the one below that is the best fit for your research. If you are not sure, read the appropriate sections before making your selection.

☐ Life sciences  ☐ Behavioural & social sciences  ☒ Ecological, evolutionary & environmental sciences

For a reference copy of the document with all sections, see [nature.com/documents/nr-reporting-summary-flat.pdf]()

# Ecological, evolutionary & environmental sciences study design

All studies must disclose on these points even when the disclosure is negative.

| | |
|---|---|
| Study description | We assembled information on song features and social behaviors for Oscine songbirds and performed phylogenetic comparative analyses. |
| Research sample | We collected information from previous studies of song, sociality, territoriality, and breeding characteristics in Oscine songbirds |
| Sampling strategy | We report sample sizes from the individual studies collected for our species-level dataset |
| Data collection | We describe our data collection in detail in the Methods; in brief, we defined search terms and inclusion criteria for published accounts of song features and social and breeding characteristics |
| Timing and spatial scale | We compiled studies at all scales of time and space |
| Data exclusions | When we compiled our dataset, we defined our criteria for excluding studies in advance and describe it in the methods |
| Reproducibility | We provide all data and code so that other researchers can reproduce our analyses or conduct their own. |
| Randomization | We compared the values of song features and social/breeding system to many randomized permutations of the species values. |
| Blinding | We collected information on song features while blind to the social and breeding system of the species. |

Did the study involve field work?  ☐ Yes  ☒ No

# Reporting for specific materials, systems and methods

We require information from authors about some types of materials, experimental systems and methods used in many studies. Here, indicate whether each material, system or method listed is relevant to your study. If you are not sure if a list item applies to your research, read the appropriate section before selecting a response.

## Materials & experimental systems

| n/a | Involved in the study |
|-----|----------------------|
| ☒ ☐ | Antibodies |
| ☒ ☐ | Eukaryotic cell lines |
| ☒ ☐ | Palaeontology and archaeology |
| ☒ ☐ | Animals and other organisms |
| ☒ ☐ | Clinical data |
| ☒ ☐ | Dual use research of concern |
| ☒ ☐ | Plants |

## Methods

| n/a | Involved in the study |
|-----|----------------------|
| ☒ ☐ | ChIP-seq |
| ☒ ☐ | Flow cytometry |
| ☒ ☐ | MRI-based neuroimaging |

## Plants

| Seed stocks | n/a |
|-------------|-----|
| Novel plant genotypes | n/a |
| Authentication | n/a |

