## [Peer Review file · Nature Ecology & Evolution]

Territoriality modulates the coevolution of cooperative breeding and female song in songbirds

Corresponding Author: Dr Nicole Creanza

Version 0:

Decision Letter:

11th July 2024

Dear Dr Creanza,

Thank you very much for your enquiry about submitting your manuscript "Cooperative breeding in songbirds promotes female song but slows the evolution of male song elaboration" to Nature Ecology & Evolution. It certainly sounds interesting, and we would be happy to consider it for publication. However, I'm sure you'll understand that we cannot make a firm decision about whether to send the paper out to review until we have carefully read the full paper (and looked at the appropriate background literature).

In order to submit your complete manuscript to Nature Ecology & Evolution, please use the link below:

Link Redacted

If you have any questions, please feel free to contact me.

Yours sincerely,

[redacted]

Version 1:

Decision Letter:

1st November 2024

Dear Nicole,

Your Article, "Cooperative breeding in songbirds promotes female song but slows the evolution of male song elaboration" has now been seen by 3 reviewers. You will see from their comments copied below that while they find your work of considerable potential interest, they have raised quite substantial concerns that must be addressed. In light of these comments, we cannot accept the manuscript for publication, but would be very interested in considering a revised version that addresses these serious concerns.

In particular, we are concerned about Reviewer #1's comments regarding data biases and definitions of mating systems (echoed by the other reviewers). We agree with the reviewer that data biases are a fundamental issue that must be resolved before we can consider a revised manuscript -- even one that addresses all other technical concerns. We of course would need to see your responses to the other referee comments as well, including additional analyses, moderation of interpretations of causality, and other textual revisions.

We hope you will find the reviewers' comments useful as you decide how to proceed. If you wish to submit a substantially revised manuscript, please bear in mind that we will be reluctant to approach the reviewers again in the absence of major revisions.

If you choose to revise your manuscript taking into account all reviewer and editor comments, please highlight all changes in

the manuscript text file [OPTIONAL: in Microsoft Word format].

* Include a "Response to reviewers" document detailing, point-by-point, how you addressed each referee comment. If no action was taken to address a point, you must provide a compelling argument. This response will be sent back to the referees along with the revised manuscript.

* If you have not done so already we suggest that you begin to revise your manuscript so that it conforms to our Article format instructions at <http://www.nature.com/natecolevol/info/final-submission>. Refer also to any guidelines provided in this letter.

Link Redacted

If you wish to submit a suitably revised manuscript we would hope to receive it within 6 months. If you cannot send it within this time, please let us know. We will be happy to consider your revision so long as nothing similar has been accepted for publication at Nature Ecology & Evolution or published elsewhere.

Nature Ecology & Evolution is committed to improving transparency in authorship. As part of our efforts in this direction, we are now requesting that all authors identified as 'corresponding author' on published papers create and link their Open Researcher and Contributor Identifier (ORCID) with their account on the Manuscript Tracking System (MTS), prior to acceptance. This applies to primary research papers only. ORCID helps the scientific community achieve unambiguous attribution of all scholarly contributions. You can create and link your ORCID from the home page of the MTS by clicking on 'Modify my Springer Nature account'. For more information please visit please visit www.springernature.com/orcid.

Thank you for the opportunity to review your work.

Yours sincerely,

[redacted]

Reviewer expertise:

Reviewer #1: mating behaviour, parental care in birds

Reviewer #2: phylogenetics, trait evolution in birds

Reviewer #3: evolution of song

Reviewers' comments:

Reviewer #1 (Remarks to the Author):

This manuscript seeks to use phylogenetic methods to seek causative associations between female song and various features of the social life of a suite of avian species. More specifically, the authors suggest that cooperative breeding may be particularly conducive to the evolution of female song.

I am afraid I was not really satisfied with the presentation of the methods used to conduct the analysis, and it is very difficult to find data in the manuscript to understand the true sample sizes on which the conclusions are based. Clear definitions are absent, and although the authors seem aware of some of the limitations, there is no interpretation of these biases. In particular, I am worried that the directionality of the hypothesis tested may be vulnerable to these biases.

First, the very low frequency of female song that has been reported in Holarctic birds seems real enough, as does the exceptionally low frequency of cooperative breeding. Because most academic ornithologists are located in the Holarctic, both of these patterns have been interpreted as 'normal', and the evolutionary direction has been interpreted as away from

this norm. However, there have been several recent studies that suggest the opposite is true for these and other traits. For example, Odom et al have suggested that female song is the ancestral trait in songbirds, and we need to explain why it has been lost in some species. Starting with Ekman's papers on corvids, and followed with reviews by Cockburn (see in particular *Adv Study of Behaviour*), the rarity of cooperative breeding in the Holarctic has also recently been re-interpreted as a loss as species recolonised postglacial landscapes. The same is also true of many other avian traits eg in several clades, in tropical species both males and females are brightly coloured, and the dull plumage of females is a derived trait at more extreme latitudes.

So what might be the biases of the data sources used. Consider the preconception that female song is generally rare. In particular, it has sometimes been estimated that it is only possible to tell the sexes apart from observations in the wild in about half of bird species. It has also been argued that cooperatively breeding birds are more likely to be sexually monomorphic and monochromatic than other species. These factors clearly hinder detection of sex bias in song, even in the tropics and south temperate areas where song is less likely to be assumed than in the Palearctic where this bias originated. It is easier to determine if the birds have been ringed and sexed via molecular or morphometric means, but such detail is most likely to be available for birds which have a social system that is uninterpretable without colour-ringing, such as cooperative breeding. However, the number of such species in the tropics is only a tiny fraction of the species of songbirds – for canopy birds such data are almost non-existent.

The same goes for the various forms of social organisation. Cooperative breeding is often (though not universally) associated with year-round occupancy of territories, and is thought to be associated with long lifespans, so these territories can be defended over many years. This imposes stochastic but rare opportunities for young birds to gain vacancies, which favours queuing for vacancies rather than early natal dispersal. This in turn sets the scene for helping behaviour to evolve. Once again, the demonstration of these behaviours (and which came first) is a moot point without colour-ringing. However, because colour-ringing is most likely when the social system is complex, cooperative breeders are likely to be over-represented.

Therefore, if we actually want to gain quantitative insights, the only tractable presentation of the data is the small Table on the top-left corner of Figure 2, but we don't know where the species came from etc, or how different this would look if other causative metrics (year-round territoriality, lifespan etc) would paint a different picture. Nor are these alternatives adequately defined. For example, 'family living' could include birds where offspring spend the non-breeding season with the parents and then disperse, spend their first breeding season with the parents but do not help etc. It is obvious that the authors could verify data better for Holarctic species see eg lines 403-406, particularly as the Birds of the World data are derived from the forerunner from North America.

Reiterating the Holarctic/rest of the world dichotomy will indubitably lead to a narrow focus on that contrast. However, there are virtually no large clades of birds where cooperation is ubiquitous, and explanations of the differences within-clade diversity are often unconvincing. There do appear to be some clades where female song seems very common, but that is based on assumptions.

Reviewer #2 (Remarks to the Author):

Summary: This manuscript describes phylogenetically controlled analyses that seek to understand the evolution of female song and the evolution of male song repertoires (in addition to some other things). The manuscript is well written and overall well presented. I was somewhat convinced by the claim that cooperative breeding and female song are linked, however there is a published pre-print (mentioned in next paragraph) that does not find similar evidence using somewhat similar but perhaps more robust phylogenetic analyses with lots of species. I suspect that having multiple evolutionarily correlated predictors makes this inference tricky in both manuscripts. In this manuscript, I was not convinced about the evolution of song repertoires because I do not believe there is enough consistency in the estimation of song repertoires for these values to be truly meaningful in comparative studies.

Major comments:

It seems like a recent major work is missing from the citations here – Odom et al "Sex differences in singing behaviour are predicted by territoriality and biparental care in songbirds". I see that one is still in preprint, and also under consideration at a Nature journal, so perhaps this is complicated. It also looks like their phylogenetic path analysis does not find as big a role for cooperative breeding in the evolution of female song as in this manuscript. I feel kind of dubious about phylogenetic path analysis, but I think their results shed light on the issue that there are correlated predictor variables here (like territoriality and cooperative breeding) that make it really tricky to tease out which variables are most responsible for driving the evolution of female song and its loss.

Overall, in the paper I'm more convinced by the evidence for the association between female song and cooperative breeding than between male song repertoire size evolution and cooperative breeding (except for the caveats in the prior paragraph). I have a great deal of doubt about the repertoire size analyses. Repertoires are really difficult to estimate robustly within a single species because of the challenge of following individuals in the wild, and it is my impression from having tried to do such work across species that different researchers are likely to arrive at different conclusions unless they have used extremely similar methodologies, and similar samples. In looking through the provided data, I was not surprised to see wildly different estimates for the same species. The authors have taken the median across estimates within species, which makes

sense, but I think the huge disparities in the estimates call into question whether these values represent something real. I don't think they're apples to apples across species because of the challenge inherent in making these estimates, and I'm not sure that repertoires are homologous either.

I think the discrete trait phylogenetic analyses (the Huelsenbeck et al 2003 type analyses) presented are probably robust to the issues described in Maddison and Fitzjohn 2015 because there are likely so many origins of cooperative breeding among passerines, and also (re-)evolutions of female song. But this would be worth spelling out explicitly because the methods applied are not themselves robust to those problems.

As discussed further below, I think the manuscript makes too much of those phylogenetic analyses with a failure to reject the null hypothesis. These tests seemed to be provided in places as evidence that there is no relationship between the response and explanatory variables, whereas this is not a correct interpretation of hypothesis testing. For this reason, I think a method like phylogenetic generalized linear modeling is superior because it allows for effect size estimation.

More specific:

Abstract: To me, the dogma of male song functions is that they serve in both mate attraction and intra-sexual competition. I don't know that anyone who currently studies male song thinks that across species, its only or even its primary function is mate attraction. The way this is presented in the abstract seems like a straw man.

Lines 47-48: In some of these species, helping at nests is the form of cooperative breeding and is relatively rare. Even for famous examples of cooperative breeding like western bluebirds. A little further down this variation is perhaps brought up via reproductive skew, which is very relevant. This is important – I'm not sure I'm convinced that cooperative breeding is one thing, given how the prevalence of cooperative breeding varies among cooperatively breeding species, and probably varies based on environmental factors as well (e.g. local densities or resource availability). (I'll note that this is partly addressed further down in the discussion.)

Line 48: I'd recommend including the suborder name Passeri here after first mentioning the oscines.

Lines 53-55: I think this statement about consensus ideas on male song functions is more true than the last line of abstract that could be interpreted to mean that mate attraction is something like a current paradigm. The phrase in the last sentence of the abstract is 'traditional mate-attraction paradigm', but I think if that was traditional at some point, that tradition has long passed.

Line 61: I'd suggest an additional important citation here is either West-Eberhard 1979 or 1983

Line 59: Does social cohesion within groups include maintenance of pair bonds for species that primarily live in male-female pairs? These are among Hall's (2009) Table 2 duetting functions that have some empirical support (i.e. "preventing a partner being usurped", "signalling commitment") and where the female vocalization has the relevant function.

Line 93: Is there a way to have more than two caretakers but also not be a cooperative breeder?

Table 1: In looking at Table 1, I'm wondering how cooperative breeding was defined. Some of these associations seem to happen by definition. E.g. cooperative breeding and asociality do not seem compatible by definition. But I suppose this could depend on how cooperative breeding is defined.

Lines 159-161: This sentence, to me, makes far too much of a narrow failure to reject the null hypothesis. It also makes me wonder about why alternate phylogenetic analyses that yield coefficient estimates for association – like phylogenetic generalized linear models - weren't used. They would allow effect size estimates for the strength of association, which might provide clarity about how to interpret the data. The analyses and interpretations feel very reliant on alpha here, and possibly on an interpretation of $P > \alpha$ that is equivalent to confirming the null instead of provisionally failing to reject it. I suppose in biological terms it would be easiest for me to ask something like this – are we certain based on these analyses that there is no statistical association between familial living and female song?

Lines 215-217: What would the mechanism possibly be here?

Lines 251-254: What about female song would promote cooperative breeding?

Lines 254-256: I'd reiterate my comment above that the P-value does not give the magnitude of the association.

Lines 492-501: I'm having trouble understanding what "an ANOVA on the interaction between the expected and observed rates" would mean. I would like to see more detail on this analysis to understand what was done. On its face this sounds more like a chi-squared test than an ANOVA.

I'm also wondering how robust the methods employed here are to the concerns raised in Maddison and Fitzjohn 2015.

Reviewer #3 (Remarks to the Author):

Summary of Comments

This is a timely and important article, with other major reviews and comparative studies of female song currently coming out, pointing to the continued value of research on this previously overlooked subject. Specifically, the current article addresses head-on an association of female song and cooperative breeding, which has been observed but not studied directly. The association between cooperative breeding and female song is an interesting and neglected topic that appears to explain the evolution or persistence of female song in certain lineages of birds: cooperative behavior appears to make female song advantageous and female song may in turn enable social interactions that are beneficial to cooperative breeding. The statistical analysis and approach is sound.

I have a few major comments:

1. I recommend tightening and streamlining the discussion to make the valuable and interesting new main points that you make there clear.
2. The authors place a lot of weight on directionality / causality of the transition rate results. Be careful not to over extend what can be interpreted from these results. To me, female song enabling transitions to cooperative breeding vs cooperative breeding promoting female song seem likely equally likely possibilities – each of which could occur depending on the circumstances or ancestral traits for specific lineages.
3. Similarly, the authors spend a lot of time discussing how female song leads to/enables cooperative breeding and vice versa. Alternatively, they could both be under similar selection pressures (i.e., there could be a third selective pressure selecting for both). The authors place a lot of emphasis on the evolution of female song and/or cooperative breeding leading to the other. More attention should be given to this third alternative.

Specific minor comments:

line 54: odd wording "increased recognition of female song has revealed that this framing may be incomplete". Maybe "increased recognition that female songbirds sing..."

Line 58: Good point. I like the way you state this. See this new reference for the most up to date review of female song function: <https://doi.org/10.1016/j.anbehav.2024.07.018>

Also, see this pre-print for most recent phylogenetic comparative analysis with female song:

<https://www.researchsquare.com/article/rs-4018424/v1>. Some of your results complement and others contrast these results. This makes your deep dive into the association of female song and cooperative breeding all the more interesting!

Line 65-68: this is a long, complicated sentence but makes important points. Can you break it down to make your points here clearer?

Line 81-82: "updated our previously published dataset on species-level song features encompassing song complexity and performance metrics for 339 species." I assume this dataset is predominantly male songs? Can you state something about this here or elsewhere. Also, some readers might wonder why you only categorized presence/absence for females. I presume this is because we don't have detailed information of song structure for females in most species. Make sure both of

these points are clear somewhere.

line 161: typo - "cooperatively breeding" should be "cooperative breeding"

Lines 201-221: I like the rigor with which you tested co-occurrence of traits and that you also looked at transition rate.

However, without a formal path analysis, I worry about the ability to talk about causality. Be careful not to put too much emphasis on which variable is causal in these transition analyses. I think that some of the transitions, the causal relationship is the other way around. For example, loss of bi-parental care is known to lead to the loss of female song (not vice versa). Be careful in your interpretation here. Also, can you make this point clear to the readers?

Discussion: Overall, I like your discussion, especially comparing the evidence for sexual competition, social conflict and social cohesion. However, overall I think the discussion could be more streamlined and pointed. I expected three subsequent paragraphs that clearly compared / contrasted evidence for each of these hypotheses. Instead there seems to be several paragraphs that blend into a progressive idea toward arguing for social cohesion. I think your discussion will be more impactful if it more clearly states and compares/contrasts these ideas. The discussion could also be shorter overall. Condensing the text and arguments will also help make the points clearer and more impactful.

To do this, make each topic and supporting evidence for each concept clearer. Also, social selection, while now being more accepted, is still a newer concept. Be clear what you mean by social competition when you first discuss it.

Line 252-254: I agree that there is a clear association between female song and cooperative breeding. However, I think it is difficult with your analyses to be certain of the causality of the relationship. I can also easily see how (and it actually seems even more likely to me) that cooperative breeding promotes female song. I suggest removing this causal statement from the first sentence of the discussion.

Line 264: Your analyses and comparisons are within the context of your association of cooperative breeding and female song. Be careful to keep this in the forefront and make sure your readers are aware that your discussion and results are specific to this context and association. For example, the specific statement "However, this predicted trend may be weak in Oscine songbirds" is only true for your associated data of cooperative breeding and female song. In fact, in broader analyses show that intermediate levels of female song is associated with polygynous mating systems. Also, it is not clear what you mean by the predicted trend. Restate this to be clear that this is specifically within the context of cooperative breeding and state what the predictions are.

Line 274: I did not follow the logic here that this is "consistent with cooperative breeding reducing sexual selection pressures on males and/or increasing them on females." Isn't it consistent with cooperative breeding reducing sexual selection pressures on males and females? Since female song is more stable in monogamous species? I'm not sure I agree that females face more competition in monogamous systems. Also, males often provide direct benefits in polygynous systems like providing food for the mate and offspring. I would think that competition is even between males and females in monogamous systems.

Line 276-282: The main points of this paragraph could be clearer.

Line 283: This is a pivotal paragraph. Clearly state what this paragraph is about at the start (what hypothesis are you investigating / supporting here?).

Line 284-285: I am not sure what you mean by negative and positive interactions. It might not be negative for the bird, if it wins them a territory or mate? Is this the best wording? What about competitive vs affiliative or other neutral language?

Version 2:

Decision Letter:

4th November 2025

Dear Nicole,

Thank you for submitting your revised manuscript "Territoriality modulates the coevolution of cooperative breeding and female song in songbirds" (NATECOLEVOL-24071861B). It has now been seen again by one of the original reviewers and their comments are below. The reviewer finds that the paper has improved in revision, and therefore we'll be happy in principle to publish it in Nature Ecology & Evolution, pending minor revisions to comply with our editorial and formatting guidelines.

If you have not done so already, please ensure that you also email us a completed copy of the Reporting summary :

Reporting summary: https://www.nature.com/documents/nr-reporting-summary.pdf

Sincerely,

[redacted]

Reviewer #1 (Remarks to the Author):

The referees have all raised aspects of two issues with the original manuscript. The first of these is the degree to which data incompleteness can affect the conclusions, and the second is the ability of comparative methods such as those used to use in this paper to resolve the correlation versus causations problem.

With respect to the first issue I think the authors have done as comprehensive a job as is currently possible to highlight and deal with issues of data unevenness.

With respect to the correlation-causation conundrum, I think there is a range of opinions among comparative biologists on the utility of phylogenetic regressions and path analyses. I have to say that I am reasonably well known as being among the sceptics of the ability of the techniques, but my scepticism has not stopped these techniques being widely implemented and refined. I do not think my scepticism warrants holding up the manuscript any further. I congratulate the author for the comprehensive response to the referees comments

Reviewer #1 (Remarks on code availability):

I have not run the code but it seems that everything is annotated. In response to the comments of all of the referees the authors have provided large numbers of additional analyses in this version. The dataset remains the same. The methodological changes in this version are well portrayed,

Reviewers' comments:

Reviewer #1 (Remarks to the Author):

This manuscript seeks to use phylogenetic methods to seek causative associations between female song and various features of the social life of a suite of avian species. More specifically, the authors suggest that cooperative breeding may be particularly conducive to the evolution of female song.

I am afraid I was not really satisfied with the presentation of the methods used to conduct the analysis, and it is very difficult to find data in the manuscript to understand the true sample sizes on which the conclusions are based.

Thank you for your careful reading of our manuscript! We have addressed all of the points that you raised in your review and substantially edited the manuscript to reflect these additional important considerations. We edited the methods section of the paper for clarity, and, to complement the sample sizes that were previously only in the Supplemental information, we now present the sample sizes used in each experiment alongside the results of that experiment. More detailed sample sizes are also available in Supplemental tables S1-S3 & S12.

Example of including sample size when describing each analysis (e.g. Line 199, but we made this change throughout the manuscript):

“Fig. 3: Coevolution and correlated evolution of female song presence and cooperative breeding. A) Overlap of stochastic character map simulations of female song and cooperative breeding, measured as the proportion of the tree occupied by each state combination (empirical p-value 0.002, 500 simulations, n = 1041 species)...”

Clear definitions are absent[...]

Thank you for noting the need for clearer definitions. We have clarified the definitions of the most relevant traits in the Methods section, and we defined each specific character state in Supplemental Table 2. The methods section on “Compiling the database” (beginning on line 424) has been divided into subsections so the definitions of different traits are easy to find.

[...][and although the authors seem aware of some of the limitations, there is no interpretation of these biases. In particular, I am worried that the directionality of the hypothesis tested may be vulnerable to these biases. First, the very low frequency of female song that has been reported in Holarctic birds seems real enough, as does the exceptionally low frequency of cooperative breeding. Because most academic ornithologists are located in the Holarctic, both of these patterns have been interpreted as ‘normal’, and the evolutionary direction has been interpreted as away from this norm. However, there have been several recent studies that suggest the opposite is true for these and other traits. For example, Odom et al have suggested that female song is the ancestral trait in songbirds, and we need to explain why it has been lost in some

species. Starting with Ekman's papers on corvids, and followed with reviews by Cockburn (see in particular Adv Study of Behaviour), the rarity of cooperative breeding in the Holarctic has also recently been re-interpreted as a loss as species recolonised postglacial landscapes. The same is also true of many other avian traits eg in several clades, in tropical species both males and females are brightly coloured, and the dull plumage of females is a derived trait at more extreme latitudes.

So what might be the biases of the data sources used. Consider the preconception that female song is generally rare. In particular, it has sometimes been estimated that it is only possible to tell the sexes apart from observations in the wild in about half of bird species. It has also been argued that **cooperatively breeding birds are more likely to be sexually monomorphic and monochromatic** than other species. These factors clearly hinder detection of sex bias in song, even in the **tropics and south temperate areas where song is less likely to be assumed than in the Palearctic where this bias originated**. It is easier to determine if the birds have been ringed and sexed via molecular or morphometric means, but such detail is most likely to be available for birds which have a social system that is uninterpretable without colour-ringing, such as cooperative breeding. However, the number of such species in the tropics is only a tiny fraction of the species of songbirds – for canopy birds such data are almost non-existent.

The same goes for the various forms of social organisation. Cooperative breeding is often (though not universally) associated with **year-round occupancy of territories**, and is thought to be associated with **long lifespans**, so these territories can be defended over many years. This imposes stochastic but rare opportunities for young birds to gain vacancies, which favours queuing for vacancies rather than early natal dispersal. This in turn sets the scene for helping behaviour to evolve. Once again, the demonstration of these behaviours (and which came first) is a moot point without colour-ringing. However, because colour-ringing is most likely when the social system is complex, **cooperative breeders are likely to be over-represented**.

We greatly appreciate you raising these crucial points. We have grouped several paragraphs of your review to respond to here because we addressed the multiple potential biases that you raised by doing a deep dive into the data and running a set of new analyses.

We have performed comprehensive additional bias analyses to test whether the availability of female song and/or cooperative breeding data systematically differs across ecological, social, and geographic categories. Specifically, we tested whether, among Oscine species in our dataset, we observe evidence for the following sources of potential bias: 1) cooperative breeding species are more likely to be studied, and thus female song presence or absence is documented with a greater likelihood in these species; 2) birds that form social groups, including colonial species and polygynous species, are more likely to be studied, and this oversampling leads to a greater likelihood that the presence or absence of female song and cooperative breeding is known in these species; 3) it is more likely for complex breeding systems or female song to be detected in year-round territorial birds or birds that exhibit strongly defensive behaviors of territory, so cooperative breeding and female song data is oversampled in species with year-round or strong territoriality; 4) greater sexual dichromatism leads to a higher rate of

identification of female song, so species with lower sexual dichromatism would either be underrepresented in the dataset overall or would be more likely to be miscategorized as lacking female song; 5) greater sexual dimorphism leads to a higher rate of identification of female song; 6) holarctic species are overrepresented in the dataset due to the uneven distribution of researchers around the globe.

We find that data availability is significantly higher in species that are territorial year-round, that are strongly territorial, that are more sexually dimorphic, and exhibit greater sexual dichromatism — traits that likely facilitate long-term study and simplify sex identification. Furthermore, female song data, but not cooperative breeding data, are more likely to be available in Holarctic species (particularly non-cooperative species), reflecting regional disparities in research effort. When we examined the interaction between geography and cooperative breeding, we found that non-cooperative breeders were more likely to have female song data in the Holarctic than in tropical regions and cooperative breeders were more likely to have female song data in the tropical than the holarctic regions (Supplemental Table 13).

To account for these biases in data sampling, we performed downsampling analyses, removing a randomly chosen subset of species from each oversampled group 500 times and re-running the phylogenetic path analyses (Methods lines 686-713), and we found that our main conclusions remain robust (Figure 4, Figure ED7, Supplemental Table 14). To summarize, our assessment showed evidence consistent with data sampling biases — (1) female song presence or absence has been better studied in temperate regions, (2) female song presence or absence appears to be easier to detect in dichromatic and dimorphic species and in territorial species (3) female song might be slightly (but not significantly) better studied in cooperative breeding species; however, **we did not detect any evidence that these sampling biases have led to a spurious association between cooperative breeding and female song in our analyses.**

To conduct these assessments of sampling bias, we added several variables to our dataset, including territoriality, body size, plumage dichromatism, size dimorphism, and geographic region. In response to reviewer comments later in this document, we conducted additional analyses using phylogenetic generalized linear modeling in which we tested whether including these factors produced a better model, and if so, how their inclusion affected the core relationships between cooperative breeding and female song.

In particular, we appreciated the reviews prompting us to consider the common effect of territoriality on cooperative breeding and female song: do we see an association between cooperative breeding and female song because both of these traits are strongly and positively associated with a third variable, namely territoriality? We used the territoriality data from Tobias *et al.* (2016) to incorporate two classifications of territoriality into our analyses: year-round territoriality and strength of territoriality. We chose to test both of these classification methods because we hypothesized that agonistic behaviors related to territoriality (i.e. strength or intensity of territorial behaviors) may be just as or more important of a modulator of the evolutionary pressures on female song as the temporal axis of territorial behavior, and since we

noted that the year-round classification method grouped species that seasonally aggressively defend territories alongside species with relatively weak territoriality. Detailed methods for the strength of territoriality classification are provided in the Methods (lines 466-491), with the specific source quotes for each species we evaluated provided in Dataset S4, but, briefly, we used Birds of the World to classify the 523 species for which we had both female song and cooperative breeding classifications and a “weak/seasonal territoriality” classification from Tobias *et al.* (2016) as either exhibiting “weak” territorial behavior or “strong” territorial behavior. We were able to make a definitive determination for 357 of these 523 species, for a final set of 875 species with classifications for female song, cooperative breeding, and territoriality strength, complementing our dataset of 1041 species with female song, cooperative breeding, and year-round territoriality classifications. We performed all analyses—state co-occurrence, state transition, phylopath, and phyloglm—using each of these territoriality classification schemes.

Table now included in the Supplemental Information as Supplemental Table 12:

			Non-cooperative	Cooperative
A	Weak or no territoriality	Female Song Absent	101	3
		Female Song Present	56	10
	Strong territoriality	Female Song Absent	172	20
		Female Song Present	425	88
B	Weak, seasonal, or no territoriality	Female Song Absent	322	15
		Female Song Present	277	23
	Year-round territoriality	Female Song Absent	36	9
		Female Song Present	281	78

Repeating our analyses using these alternative territoriality classifications provided new insights into the coevolutionary dynamics between female song, cooperative breeding, and territorial behavior. Not only do we find that our original main finding—a robust association between female song and cooperative breeding—persists when controlling for either measure of territoriality, we also find additional support for our original theoretical interpretation of this link, namely, a potential social-cohesion function of female song that increases fitness in cooperative lineages; although cooperative breeding and female song are both more common in strongly territorial species, the predictive power of cooperative breeding on female song is strongest in weakly territorial contexts, where the two traits co-occur much more than expected by chance (Supplemental Table 12).

Some additions to the manuscript and supplement to test the effects of data biases:

Supplemental Table 13: Tests for sampling biases in female song and cooperative breeding data availability. We tested whether data availability for female song presence/absence and cooperative breeding classification was biased with respect to various species traits that could confound evolutionary analyses. For each bias type, we tested the null hypothesis that data availability was independent of the trait in question. Chi-squared tests were used for categorical traits and Welch's t-tests for continuous traits. Significant biases ($p < 0.05$) indicate that species with certain trait values are over- or under-represented in our dataset. Geographic bias was tested using biogeographic regions from Jetz & Rubenstein (2011), with Holarctic (Nearctic + Palearctic) compared to tropical regions. Sexual dichromatism values were obtained from Dale et al. (2015) and sexual dimorphism from wing measurements in AVONET (Tobias et al. 2022). These detected biases were addressed through downsampling procedures (Table S14) to ensure the robustness of our main findings.

Bias Type	Hypothesis	Test	Test statistic	df	p-value	Result
Cooperative breeding	Cooperative breeding species more likely to have female song data	Chi-squared	$\chi^2 = 2.18$	1	0.14	No bias in FS data
Coloniality	Colonial species more likely to be studied	Chi-squared Chi-squared	$\chi^2 = 0.01$ $\chi^2 = 2.06$	1 1	0.91 0.15	No bias in FS data No bias in CB data
Polygyny	Polygynous species more likely to be studied	Chi-squared Chi-squared	$\chi^2 = 2.143$ $\chi^2 = 0.46$	1 1	0.143 0.498	No bias in FS data No bias in CB data
Geographic	Holarctic species more likely to be studied	Chi-squared Chi-squared	$\chi^2 = 56.80$ $\chi^2 = 0.74$	1 1	<0.001 0.785	Bias detected in FS data No bias in CB data
Geographic x Cooperative breeding	Female song may be more or less likely to be studied based on the interaction between their geographic region and cooperative breeding status	Chi-squared	$\chi^2 = 56.92$	3	<0.001	Bias detected in FS data (Post-hoc tests found that non-cooperative holarctic species and cooperative tropical species are both overrepresented in the female song data)
Territoriality (Weak/Strong)	More territorial more likely to be studied	Chi-squared Chi-squared	$\chi^2 = 58.67$ $\chi^2 = 31.49$	1 1	<0.001 <0.001	Bias detected in FS data Bias detected in CB data
Territory (Year-round)	Year-round territorial more likely to be studied	Chi-squared Chi-squared	$\chi^2 = 89.22$ $\chi^2 = 39.18$	1 1	<0.001 <0.001	Bias detected in FS data Bias detected in CB data
Sexual dichromatism	Dichromatic species more likely to have FS data	t-test	t = 3.46	1789	<0.001	Bias detected in FS data

Sexual dimorphism	Dimorphic species more likely to have FS data	t-test	$t = -2.77$	1759	0.006	Bias detected in FS data
-------------------	---	--------	-------------	------	-------	--------------------------

Figure 4:

Fig. 4: The links between cooperative breeding and female song in phylogenetic path analyses are robust to correcting for data availability biases. (A) Directed acyclic graph showing conditional-averaged path coefficients from phylogenetic path analysis of the relationships between female song presence, cooperative breeding, strength of territoriality, and body mass (log-scaled and normalized). Arrow thickness indicates the magnitude of path coefficients, with all paths shown representing positive relationships. Only models within $\Delta\text{CICc} < 2$ of the best model (4 models) were included in the averaging. (B) Forest plot showing the mean and 95% confidence interval for the cooperative breeding → female song path coefficient across 500 iterations for each bias correction type, compared to the value from the full dataset (dashed line). (C) Forest plot showing the mean and 95% confidence interval for the female song → cooperative breeding path coefficient across 500 iterations for each bias correction type, compared to the value from the full dataset. (D) Model consistency across bias corrections. Heatmap showing the number of iterations for which each phylogenetic path model was significant (had $\Delta\text{CICc} < 2$) across 500 iterations for each bias correction type. The row underneath the heatmap contains the ΔCICc values for each model in the phylopath analysis performed using the full dataset, with the ΔCICc values of the four models that were statistically equivalent ($\Delta\text{CICc} < 2$) in the full dataset in italics. The lower panel shows the models based on the presence or absence of each rate in the model. The models shown are

all those that were in the top 20 models (out of 47 total models, see Fig. S1) by # of iterations with $\Delta\text{CICc} < 2$ for any of the bias correction types.

Therefore, if we actually want to gain quantitative insights, the only tractable presentation of the data is the small Table on the top-left corner of Figure 2, but we don't know where the species came from etc, or how different this would look if other causative metrics (**year-round territoriality, lifespan** etc) would paint a different picture.

We are glad that you shared our intuition that the raw numbers of species with female song and cooperative breeding (Figure 2) gave a first hint that the two features might have an underlying association. We now include several additional analyses where we consider other metrics alongside these focal traits, which we think help alleviate the concern that there is another causative metric and/or a relationship with a confounding variable that leads to a spurious association between cooperative breeding and female song. First, we address the concern that another variable, such as territoriality, is associated with both cooperative breeding and female song, making them appear more linked with one another than they are. In our new analyses of cooperative breeding and female song, we now consider each of two measures of territoriality, (a) year-round territoriality vs. seasonal or no territoriality or (b) strong vs. weak or no territoriality, and we find that cooperative breeding and female song are associated with one another even when we account for territoriality. Lifespan data were only available for a small subset of species, so we used body mass as a proxy (and we note that body mass itself was positively associated with female song in the recent Odom *et al.* 2025 paper, and with cooperative breeding—see our Supplemental Table 20); again, cooperative breeding and female song are associated with one another beyond what would be predicted by their relationship to body size. In addition, we considered the possibility that a social variable related to cooperative breeding is a better predictor of female song than cooperative breeding itself; none of the other social variables we tested were significantly stronger predictors of female song except for number of caretakers, which is essentially a numerical representation of the cooperative breeding categorization (i.e. species with 1 or 2 caretakers make up the “non-cooperative” category and species with 3 or more caretakers make up the “cooperative” category). To address your point about “where the species came from,” we also performed analyses that included latitude of the species range centroid and whether the species was classified as holarctic vs. tropical; these geographic features also did not account for the relationship between cooperative breeding and female song (Supplemental Table 23; species counts provided in Supplemental Table 25, included below).

Geographic Region	Cooperative Breeding	Female Song	
		Absent	Present
Holarctic	non-cooperative	69	140
Tropical	non-cooperative	280	412
Holarctic	cooperative	3	8
Tropical	cooperative	20	92

Nor are these alternatives adequately defined. For example, 'family living' could include birds where offspring spend the non-breeding season with the parents and then disperse, spend their first breeding season with the parents but do not help etc. It is obvious that the authors could verify data better for Holarctic species see eg lines 403-406, particularly as the Birds of the World data are derived from the forerunner from North America.

We agree that there is inherently a lot of nuance that gets lost when converting a set of complex behaviors into a numeric vector. However, large-scale comparative analyses of behaviors are only possible when we recognize the common threads that exist between species' behaviors and then categorize or quantify these behaviors based on clear criteria. For most variables, we used published datasets and thus now include the definitions of those variables from their original sources. For example, we use a definition of "family living" from Griesser *et al.* 2017 (the source from which we obtained the family living data), in which a species is considered to have family living present (for a binary classification) if their offspring remain with the parents beyond nutritional independence, which is further subdivided (for a multi-state classification) into those species whose offspring help in rearing versus those species whose offspring do not help with rearing in the original source (Griesser *et al.* 2017 variable "social_system", with groups "fam", "coop_fam", and "no_fam"). Following these definitions, the two examples you give could be reliably classified into a category, even by different researchers. For the key female song and cooperative breeding variables, we prioritized maintaining consistency with definitions used in other published work, and verified and added additional data following the criteria outlined in the Methods of this paper (Lines 425-513).

As we described above, your intuition that Holarctic birds would be oversampled was supported by our analysis to detect biases in the data sampling. However, downsampling Holarctic species did not eliminate our ability to detect an association between cooperative breeding and female song.

Reiterating the Holarctic/rest of the world dichotomy will indubitably lead to a narrow focus on that contrast. However, there are virtually no large clades of birds where cooperation is ubiquitous, and explanations of the differences within-clade diversity are often unconvincing. There do appear to be some clades where female song seems very common, but that is based on assumptions.

We appreciate the valid concerns that you raised. We have taken these points to heart, gathered additional necessary data, and broadened our analyses to both test and account for these data biases. We have now performed extensive bias testing and, where significant data availability biases exist (namely, that female song data is overrepresented based on geographic region, cooperative breeding, territoriality, plumage dichromatism, and size dimorphism), we performed repeated phylogenetic path analyses with systematic downsampling of species. We found that these data corrections did not change the conclusions that we draw based on the results of our full-dataset analyses. We also now include territoriality and body size in all

multi-factorial analyses performed. We believe that this additional context has made our study significantly stronger and our conclusions both more confident and more nuanced.

Reviewer #2 (Remarks to the Author):

Summary: This manuscript describes phylogenetically controlled analyses that seek to understand the evolution of female song and the evolution of male song repertoires (in addition to some other things). The manuscript is well written and overall well presented. I was somewhat convinced by the claim that cooperative breeding and female song are linked, however there is a published pre-print (mentioned in next paragraph) that does not find similar evidence using somewhat similar but perhaps more robust phylogenetic analyses with lots of species. I suspect that having multiple evolutionarily correlated predictors makes this inference tricky in both manuscripts. In this manuscript, I was not convinced about the evolution of song repertoires because I do not believe there is enough consistency in the estimation of song repertoires for these values to be truly meaningful in comparative studies.

Thank you for your thorough reading! We appreciate your careful feedback, and we address all of the important points mentioned in your summary paragraph in the Major comment where they are described.

Major comments:

It seems like a recent major work is missing from the citations here – Odom et al “Sex differences in singing behaviour are predicted by territoriality and biparental care in songbirds”. I see that one is still in preprint, and also under consideration at a Nature journal, so perhaps this is complicated. It also looks like their **phylogenetic path analysis** does not find as big a role for cooperative breeding in the evolution of female song as in this manuscript. I feel kind of dubious about phylogenetic path analysis, but I think their results shed light on the issue that there are correlated predictor variables here (like territoriality and cooperative breeding) that make it really tricky to tease out which variables are most responsible for driving the evolution of female song and its loss.

Thank you for noting the omission of the Odom *et al.* preprint, which we had read but not cited; their manuscript has very recently been published, and we now cite the published version. We have also now included both phylogenetic path analysis and phylogenetic generalized linear models in our methods so that we can more directly test the associations between female song and cooperative breeding. We appreciate your important point about the tricky nature of teasing apart correlated variables in evolutionary analyses, and we have tried to approach our hypothesis from multiple angles to assess whether cooperative breeding and female song are evolutionarily correlated over and above what would be expected from their mutual association with other factors.

In general, our results are largely concordant with the comparable results in the Odom *et al.* paper: we find consistent associations between female song, territoriality, body mass, and cooperative breeding across multiple analysis methods. We also use a similar sample size; they have 1010 species with female song occurrence and cooperative breeding data, and we have 1041 species with female song and cooperative breeding data. They found that cooperative breeding was included as a predictor of female song in their best models, but the effect size was relatively small so they did not delve into the interaction much. With our data and approach, we find evidence for a stronger interaction than they report, even when we account for the other factors that they consider in their models.

Since our initial hypothesis was that cooperative breeding and female song might interact, we constructed our dataset to be best able to test the potential relationship between these two variables. We assembled a cooperative breeding dataset that harmonized 10 different comprehensive studies of cooperative breeding presence and absence, and supplemented it with information from Birds of the World. We also synthesized multiple databases of female song information, and we encoded territoriality as two different meaningful binary variables instead of using a continuous metric of three ordinal categories.

Most importantly, now that their article is published and the data are available as supplementary files, we note a crucial consideration that we had not detected in the preprint. One of their main findings was that female song had an evolutionary relationship with biparental care, whereas we found a positive association between female song and larger numbers of caretakers, which suggests that there is greater predictive power when the number of caretakers is greater than two rather than equal to two. They describe their result as involving “biparental care” (e.g.: “In line with the sexual selection hypothesis (Figure 1), species with female song most often had biparental care, whereas species without biparental care usually lacked female song.”) so we had assumed biparental care in their analysis meant “care by two parents”, i.e. having two caretakers. However, in their supplemental data, this variable is called `paternal_care`, and seems to correspond to the presence of a father during caretaking even if other non-parental individuals are also helping with care. All species that are categorized as cooperative breeders in their dataset also have “`paternal_care`” present in their dataset, presumably because the father is present during the cooperative breeding (perhaps not an inherently problematic assumption), despite the fact that the primary source they cite for these data (Cockburn 2006) categorizes the caretaking strategies of “cooperative” separately from “pair”. However, to reiterate your point about it being tricky to disentangle related variables, it seems likely that cooperative breeding as its own variable has a smaller effect size in their results than in ours because its presence is also encoded in another variable in their analysis. It would be interesting to repeat their analysis with the presence of biparental care defined as “care by two parents” (as opposed to, presumably, “care by two parents either on their own or as part of a cooperative breeding group”) and assess whether their cooperative breeding variable then has greater predictive power.

Overall, in the paper I'm more convinced by the evidence for the association between female song and cooperative breeding than between male song repertoire size evolution and cooperative breeding (except for the caveats in the prior paragraph). I have a great deal of doubt about the repertoire size analyses. Repertoires are really difficult to estimate robustly within a single species because of the challenge of following individuals in the wild, and it is my impression from having tried to do such work across species that different researchers are likely to arrive at different conclusions unless they have used extremely similar methodologies, and similar samples. In looking through the provided data, I was not surprised to see wildly different estimates for the same species. The authors have taken the median across estimates within species, which makes sense, but I think the huge disparities in the estimates call into question whether these values represent something real. I don't think they're apples to apples across species because of the challenge inherent in making these estimates, and I'm not sure that repertoires are homologous either.

We appreciate your concern about the song repertoire estimates, and we thank you for raising it. We agree that different researchers might assess repertoire size differently (or might evaluate different durations of song or different numbers of individual birds, potentially leading to different estimates for a species even when using the same methods), and the individual data points that we report for different species can vary quite a bit. In addition to taking the median of these individual estimates for each species, we had also repeated our analysis with the maximum and the minimum repertoire size estimate. Essentially, this procedure allowed us to ask "if we only found the highest estimate of song repertoire for each species, would we have made the same evolutionary conclusions? What if we had only found the lowest estimate?" In general, we concluded that our methods were robust to variation in the estimates. In response to your feedback, we added another important control: we performed this test again with resampling to randomly choose the minimum, median, or maximum published value for each species, for 500 iterations of resampled values, 2 simulations per resampled vector, for 1000 total simulations (Methods lines 577-583). This result was consistent with our original findings (99.3% of simulations showed a faster rate of evolution of song repertoire size in non-cooperative lineages, p-value calculated from the mean log-likelihood across simulations 0.0332). This new test is essentially asking "If we had only found one repertoire size estimate for each species by chance, would we have made the same conclusions?" We found that each of these resampling procedures recapitulated our original results that song repertoire size evolved more slowly in cooperatively breeding species. We think that this consistency between resampling schemes is because, despite the variance in the individual estimates of song repertoire size, there is an overall phenomenon in which some species have larger song repertoires than others, and the general pattern is more important than the precise number. For example, song repertoire size estimates for the Fox sparrow range from 1 to 3.2, for the Northern cardinal range from 8 to 18.67, for the European starling range from 35 to 48, and the Northern mockingbird range from 190 to 500. Even when the highest estimate is more than double the lowest estimate for most of these species, the general ranking of which of these species have small, medium, and large song repertoires is preserved by any number within the range of estimates. While we share your view that what constitutes discrete and unique "songs" varies across species (and across

researchers), we think that broad trends likely hold a fair degree of biological validity for estimating the relative diversity of individual song repertoires across species.

I think the discrete trait phylogenetic analyses (the Huelsenbeck et al 2003 type analyses) presented are probably robust to the issues described in Maddison and Fitzjohn 2015 because there are likely so many origins of cooperative breeding among passerines, and also (re-)evolutions of female song. But this would be worth spelling out explicitly because the methods applied are not themselves robust to those problems.

Thank you for noting this important consideration with the type of analysis we conducted here! We agree that the numerous evolutionary origins of both female song and cooperative breeding make it robust to the problems with trying to conduct phylogenetic comparative analyses on traits that are clumped in one region of the tree, as described by (Maddison and FitzJohn 2015). We now draw attention to the multiple evolutionary origins of both cooperative breeding and female song in the caption of Figure 2 so that the reader can visually verify that the traits are not associated due to an effect from a single evolutionary event.

New text (in Methods lines 609-612): “We also note that the methods of (Huelsenbeck *et al.* 2003) can be misleading when one or more of the studied traits are clustered in one small section of the tree or have one major origin in evolutionary history (Maddison and FitzJohn 2015); however, when we visualize these traits on a large phylogeny, we observe many gains and losses of both female song and cooperative breeding throughout the tree (Figure 2).“

As discussed further below, I think the manuscript makes too much of those phylogenetic analyses with a failure to reject the null hypothesis. These tests seemed to be provided in places as evidence that there is no relationship between the response and explanatory variables, whereas this is not a correct interpretation of hypothesis testing. For this reason, I think a method like **phylogenetic generalized linear modeling** is superior because it allows for effect size estimation.

We agree with this important point about not overinterpreting failures to reject null hypotheses. We have addressed this by 1) omitting or modifying statements that incorrectly claimed a non-significant association between two factors as evidence to conclude that those factors were not related and 2) performing additional analyses that allow us to compare the predictive power of different factors. These include phylogenetic generalized linear models (phyloglm), which we now implement and report effect size estimates in addition to our previous analyses. For our binary response variables (female song and cooperative breeding), we tested models with different combinations of predictors (cooperative breeding, female song, territoriality, body mass) and their interactions, using AIC for model selection. These analyses revealed strong effect sizes for the cooperative breeding-female song association (cooperative breeding predicting female song: Odds ratio = 5.70; female song predicting cooperative breeding: Odds ratio = 6.88). (Lines 238-247)

We also used phyloglm to address the question we were previously inappropriately attempting to answer by referring to our non-significant results, which you rightly critiqued: are any social variables, especially those associated with cooperative breeding, better predictors of female song than cooperative breeding? To explicitly address this question, we performed direct model comparisons. For social variables that co-occurred with cooperative breeding in evolutionary history (Supplemental Table 10)—colonial living, familial living, group size, social bond duration, and number of caretakers—we compared models including that variable as a predictor versus models including cooperative breeding as a predictor (alongside territoriality and body mass), applied to the subset of species that had data for both cooperative breeding and the social variable in question. We note that this subsetting reduced the numbers of species that could be included in the model by over 4-fold for some social features. Nevertheless, across most comparisons, models including cooperative breeding consistently outperformed or matched models with alternative social predictors (Table 2). Cooperative breeding showed significant and substantially stronger predictive power than colonial living, group size, and social bond duration, and stronger predictive power than familial living, though neither were significant in their common subset of species. Only the number of caretakers showed stronger predictive power than binary cooperative breeding, which is expected given that it is essentially a numerical representation of the binary cooperative breeding vector (with non-cooperative species having 0, 1, or 2 caretakers and cooperative species having more than two caretakers). These analyses demonstrate that the cooperative breeding-female song association cannot be explained by these other social variables.

Additionally, we performed stepwise model selection by iteratively adding potential confounding variables (familial living, sexual dimorphism, geographic factors, migration, etc.) to determine which factors significantly improved model fit ($\Delta AIC > 2$). While familial living emerged as a strong predictor of cooperative breeding (OR = 14.61; CI = 5.28-35.33, $p < 0.002$), it did not improve the model to predict female song presence when included alongside cooperative breeding. Sexual dimorphism metrics had some predictive power for female song (wing dimorphism OR = 0.83; plumage dichromatism OR = 0.85), but, critically, the cooperative breeding-female song association remained robust even when accounting for these factors (CB OR = 5.97; $p = 0.004$ in the expanded model). This demonstrates that the relationship between cooperative breeding and female song cannot be explained by their mutual associations with other life history traits (Supplemental Table 21).

More specific:

Abstract: To me, the dogma of male song functions is that they serve in both mate attraction and intra-sexual competition. I don't know that anyone who currently studies male song thinks that across species, its only or even its primary function is mate attraction. The way this is presented in the abstract seems like a straw man.

We apologize; in trimming words from the abstract we had certainly made it sound like we have a very simplified view of birdsong function. We have reworded this sentence.

Lines 47-48: In some of these species, helping at nests is the form of cooperative breeding and is relatively rare. Even for famous examples of cooperative breeding like western bluebirds. A little further down this variation is perhaps brought up via reproductive skew, which is very relevant. This is important – I’m not sure I’m convinced that cooperative breeding is one thing, given how the prevalence of cooperative breeding varies among cooperatively breeding species, and probably varies based on environmental factors as well (e.g. local densities or resource availability). (I’ll note that this is partly addressed further down in the discussion.)

We appreciate your thoughtful consideration of cooperative breeding, and we agree that one simple definition might not fully characterize the set of possible behaviors involved. For this initial definition, which is now a little later in the introduction, we have tried to rephrase to leave some room for cooperative breeding to be more than one thing. (Lines 63-65 “Cooperative breeding, which is broadly defined as a breeding system in which non-parental adults assist in rearing young, has evolved numerous times in the avian lineage, and is observed to different extents in approximately 13% of Oscine songbird species (Passeri).”)

In addition, we note that, in constructing our dataset, we made multiple versions of the cooperative breeding presence/absence vector to acknowledge that different researchers might define cooperative breeding slightly differently. Since we synthesized data from numerous sources, we had to decide what to do when a subset of studies deemed a species non-cooperative and others deemed it cooperative. For the main analyses, we considered a species to have cooperative breeding when either all sources were unanimous in categorizing it as a cooperative breeder or when we were able to resolve any conflicting responses with extra information from Birds of the World, but we repeated all analyses with the alternative cooperative breeding data vectors to ensure that results were robust to this variation. In this way, we tried to analyze species that had broad consensus on their cooperative breeding status and reduce the effect of discrepancies between definitions on our results.

Finally, we also tested a numerical variable that is closely associated with our binary categorization of cooperative breeding: number of caretakers. This variable was defined in Griesser *et al.* (2023) as follows: “mound-nesting species had zero caretakers, species with uniparental care one, species with bi-parental care two, and cooperatively breeding species the mean value of caretakers.” Thus, this variable adds resolution to the definition of cooperative breeding, where species that exhibit a mix of biparental care and cooperative breeding would have a mean number of caretakers closer to two and species that have larger cooperative groups or more frequent cooperative breeding would have greater values. When we tested this higher-resolution metric of the extent of cooperative caretaking, we found that it had even greater predictive power about the presence of female song than our binary cooperative breeding variable did (Table 2). This result suggests that the presence of female song is predicted not only by the presence of cooperative breeding in a species but also by its prevalence or extent.

Line 48: I'd recommend including the suborder name Passeri here after first mentioning the oscines.

We have added this! (Line 65)

Lines 53-55: I think this statement about consensus ideas on male song functions is more true than the last line of abstract that could be interpreted to mean that mate attraction is something like a current paradigm. The phrase in the last sentence of the abstract is 'traditional mate-attraction paradigm', but I think if that was traditional at some point, that tradition has long passed.

Thank you for noting that the introduction did a better job of describing song functions! We have rephrased the abstract and the introduction to respond to the feedback of all three reviewers, but we have retained the more nuanced characterization of male song. (e.g. Line 41-42: "Song is traditionally viewed as a sexually selected male behavior with functions related to mate attraction and territory defense")

Line 61: I'd suggest an additional important citation here is either West-Eberhard 1979 or 1983

Thank you for pointing this out! We agree that West-Eberhard has done important work in this area, and we now cite her 1983 paper here.

Reference 8, cited Line 51: "The evolution of song may thus be subject to evolutionary pressures related to social interaction in addition to traditionally studied sexual selection, a phenomenon that might be especially relevant in females^{7,8}."

Line 59: Does social cohesion within groups include maintenance of pair bonds for species that primarily live in male-female pairs? These are among Hall's (2009) Table 2 duetting functions that have some empirical support (i.e. "preventing a partner being usurped", "signalling commitment") and where the female vocalization has the relevant function.

We hypothesize that social cohesion likely does include maintenance of pair bonds, and, moreover, that social-bond maintenance vocal behaviors could effectively evolve out of these repurposed pair-bond-maintenance behaviors. The motivational systems that support pair-bond maintenance through vocal interaction, such as social attachment and anxiety reduction (described in the cited paper by Riters *et al.*), could readily be co-opted to support group cohesion in cooperative systems.

Text addressing these points:

(in Introduction, Lines 43-51) "Data on the biological contexts of female song are sparse for most species, but, broadly, several patterns of singing behavior have been observed, including male-female duetting, agonistic female-female interactions, and within-pair or within-group interactions³. These behavioral patterns correspond with various hypothesized functions of female song: mate attraction, intrasexual competition, territory defense, pair-bond maintenance,

and social cohesion within groups^{4,5}. A recent quantitative review reinforces that territory defense, intrasexual competition, and intrapair communication are the most commonly supported functions of female song, while its use in mate attraction appears rare⁶. The evolution of song may thus be subject to evolutionary pressures related to social interaction in addition to traditionally studied sexual selection, a phenomenon that might be especially relevant in females^{7,8}.”

(in Discussion, Lines 338-350) “A third hypothesis is that female song could serve a role in maintaining social cohesion in cooperative breeding groups. Song produced in the context of established breeding pairs is already well-documented as a factor in reinforcing pair bonds; however, evidence suggests that song can also mediate affiliative social interactions beyond the breeding pair, potentially by co-opting the reward systems that support pair-bond maintenance. For instance, European starlings’ “gregarious song”, which occurs outside of the breeding season, is positively correlated with social flocking behavior and is thought to be intrinsically rewarding^{36,37}. Male zebra finch “undirected song” (i.e. not directed toward a female) occurs year-round and is hypothesized to have prosocial functions, such as breeding synchronization, promoting social cohesion, and individual recognition³⁸. Female song shares some patterns with these examples (reviewed in⁴): it often occurs both in and out of the breeding season³⁹ and has been linked to breeding-related contexts including coordination of breeding activities^{4,40} and interactions with offspring post-fledging⁴¹, and non-breeding-related functions including helper retention⁴², individual or group-member recognition^{42,43}, neighbor recognition⁴⁴, and social cohesion⁴⁵.”

Line 93: Is there a way to have more than two caretakers but also not be a cooperative breeder?

This is an interesting question; in theory, based on our definitions, no. However, there is one species in our dataset, the New Holland Honeyeater (*Phylidonyris novaehollandiae*) that is reported to have a mean number of caretakers of 2.5 in the Griesser *et al.* 2023 dataset but is primarily classified as non-cooperative in our dataset due to source discrepancies, with a narrow majority of sources classifying it as “non-cooperative” (3 sources, while 2 sources classify it as “cooperative”). In *Birds of the World*, it is described as having “occasional instances of co-operative breeding, with helpers”; our criteria explicitly did not count references describing cooperative breeding as “occasional” as evidence of CB. The “2.5 caretakers” figure seems to reflect this variation in cooperative breeding in this species.

Table 1: In looking at Table 1, I’m wondering how cooperative breeding was defined. Some of these associations seem to happen by definition. E.g. cooperative breeding and asociality do not seem compatible by definition. But I suppose this could depend on how cooperative breeding is defined.

(For reference, the previous Table 1 is now Table S10.) The sociality metrics did not factor into how we defined cooperative breeding or how we assigned species to a cooperative or non-cooperative breeding strategy. In the main text, we include a conceptual definition for

cooperative breeding (Line 63: “Cooperative breeding, which is broadly defined as a breeding system in which non-parental adults assist in rearing young”), as well as a lengthy description of how we classify a species as having cooperative breeding:

Excerpt from Methods “Cooperative breeding” section, Lines 425-455: “We considered cooperative breeding behavior to be alloparenting during incubation, brooding, and nest provisioning in 10% or more of nests in a study, if quantitative data were available, following the classification methods of Cockburn³. We did not consider nest building, territory defense, or fledgling provisioning by extrapair individuals to be sufficient for cooperative breeding classification. When a source’s data were qualitative, species described unambiguously as “cooperative,” “breed cooperatively,” “often cooperative,” “facultatively cooperative breeding,” or as commonly having “helpers” at the nest were considered to be “cooperative”, and species described with “(cooperative breeding) does not occur,” “none,” “extremely rare” were considered to be “noncooperative.”

However, there are some examples of these intuitive incompatibilities present in the sociality data. In the Supplemental Information document, Table S2, we report the numbers of species in each group (e.g. acolonial vs. colonial) that have cooperative breeding and female song so that it is readily apparent which categories include essentially all cooperative breeding species. For example, there are no cooperative breeding species in our dataset that were classified as having either “asocial” or “pair” social grouping or as having “short social bonds” in the Griesser *et al.* (2023) dataset. Other variables do indeed have an association with cooperative breeding by definition, e.g. the categorization into “More than two caretakers,” since the number of caretakers was defined in Griesser *et al.* (2023) as follows: “mound-nesting species had zero caretakers, species with uniparental care one, species with bi-parental care two, and cooperatively breeding species the mean value of caretakers.”

Lines 159-161: This sentence, to me, makes far too much of a narrow failure to reject the null hypothesis. It also makes me wonder about why alternate phylogenetic analyses that yield coefficient estimates for association – like phylogenetic generalized linear models - weren’t used. They would allow effect size estimates for the strength of association, which might provide clarity about how to interpret the data. The analyses and interpretations feel very reliant on alpha here, and possibly on an interpretation of $P > \alpha$ that is equivalent to confirming the null instead of provisionally failing to reject it. I suppose in biological terms it would be easiest for me to ask something like this – are we certain based on these analyses that there is no statistical association between familial living and female song?

We agree that we worded the last sentence in this section a bit too definitively (and that the inclusion of “may” in the sentence did not hedge the statement strongly enough). We have deleted this summary sentence in favor of describing the patterns more clearly without overinterpreting them. It now reads (Lines 179-184):

“Despite the high degree of co-occurrence between familial living and cooperative breeding, the co-occurrence between familial living and female song was not significantly higher than expected by chance ($p = 0.092$, $n = 468$ species, Figure 3B, Table ED2). Performing a similar

analysis using the four-state breeding system classification dataset showed that female song occurs significantly more often in systems with kin-cooperative breeding than expected by chance and significantly less often in systems with non-kin-cooperative or non-familial, non-cooperative systems”

To directly address your question about whether we can be certain there is no association between familial living and female song, we performed phylogenetic generalized linear models that allow for effect size estimation. In direct model comparisons, cooperative breeding was a moderately stronger predictor of female song than familial living (Table 2, Supplemental Table 18). Furthermore, when both predictors were included in stepwise model selection (along with other factors such as plumage dichromatism, size dimorphism, range latitude, geographic region, and migration), familial living did not improve model fit for predicting female song, whereas it was a strong predictor of cooperative breeding (Supplemental Tables 21-24, which show that familial living was a predictor of cooperative breeding but not of female song). Therefore, while we can't definitively rule out a relationship between familial living and female song, the evidence suggests that cooperative breeding has a stronger and more consistent association with female song presence than familial living does.

Lines 215-217: What would the mechanism possibly be here?

During the revision process, the lines referenced here have been slightly edited and moved to the Supplement; they now read “Intriguingly, the presence of female song appeared to greatly reduce the likelihood that a lineage would transition to fewer than two caretakers, whereas the loss of biparental care was more likely in the absence of female song.” Since female song contributes to pair bond maintenance in many species, it follows that having this behavioral trait that reinforces pair stability could make transitions away from biparental care less likely. Species in which female song functions in this way essentially have an additional mechanism for maintaining breeding partnerships, which could create evolutionary pressure against losing biparental investment.

Lines 251-254: What about female song would promote cooperative breeding?

We have overhauled the Discussion section and now address this question more directly. The findings of both our phylogenetic path analyses and phylogenetic generalized linear modeling support a bidirectional relationship between female song and cooperative breeding, and we now discuss potential mechanisms in the context of social cohesion and group coordination. The added text reads (Lines 352-361):

“While social living may often lead to more frequent competition among females for mates or other resources, the maintenance of the social group—and the fitness benefits thereof—should select for behaviors that minimize sexual conflict, particularly in systems with both high relatedness and significant cooperation. While social bond duration and the number of caretakers also showed positive associations with female song, most other social variables

showed weak or negative associations. This pattern, where cooperative caretaking and longer social bonds promote female song, but other forms of group living do not, supports the hypothesis that female song may serve prosocial functions specifically in stable, cooperative social systems. Further, species where female song already mediates group coordination or offspring communication may be pre-adapted for transitions to cooperative breeding, as the vocal behaviors necessary for managing complex multi-caregiver systems would already be in place.”

Lines 254-256: I’d reiterate my comment above that the P-value does not give the magnitude of the association.

Thank you for noting this. We have now rewritten the discussion overall, but we have kept your point in mind. In addition, to have a better estimate of the magnitude of associations between variables, we now use phylogenetic generalized linear modeling to provide effect sizes and odds ratios.

Lines 492-501: I’m having trouble understanding what “an ANOVA on the interaction between the expected and observed rates” would mean. I would like to see more detail on this analysis to understand what was done. On its face this sounds more like a chi-squared test than an ANOVA.

Thank you for raising this point—the methods were previously written in an overly simplistic way that we agree was unclear. We actually performed the ANOVA on a linear model testing whether, across the 500 sets of simulated stochastic character maps, there was a significant difference in log-transformed state-switch counts between the observed and expected values across the 8 possible state transitions, including their interaction. We also inadvertently left out of the methods that we also performed pairwise post-hoc tests to determine which state transitions were overall different from expected across all of the simulations. We have replaced the unclear verbiage in the text with this (Lines 622-628): “We log-transformed these state-switch counts across the 500 simulations and fit a two-way linear model with count type (observed vs. expected) and state transition type (eight possible transition types per pair of binary traits) as fixed factors, including their interaction. An ANOVA on this model tested whether the difference between observed and expected counts varied among transition categories. When the interaction term was significant, we conducted Tukey-adjusted pairwise comparisons of estimated marginal means within each transition type to identify which specific transitions differed significantly from the expected counts.”

I’m also wondering how robust the methods employed here are to the concerns raised in Maddison and Fitzjohn 2015.

Thank you for reminding us of this reference; we now directly address the concerns raised by Maddison and Fitzjohn about detecting correlated evolution between two traits when one or both traits are clustered on the phylogeny.

New text (Lines 609-612):

“We also note that the methods of (Huelsenbeck *et al.* 2003) can be misleading when one or more of the studied traits are clustered in one small section of the tree or have one major origin in evolutionary history (Maddison and FitzJohn 2015); however, when we visualize these traits on a large phylogeny, we observe many gains and losses of both female song and cooperative breeding throughout the tree (Figure 2).”

Reviewer #3 (Remarks to the Author):

Summary of Comments

This is a timely and important article, with other major reviews and comparative studies of female song currently coming out, pointing to the continued value of research on this previously overlooked subject. Specifically, the current article addresses head-on an association of female song and cooperative breeding, which has been observed but not studied directly. The association between cooperative breeding and female song is an interesting and neglected topic that appears to explain the evolution or persistence of female song in certain lineages of birds: cooperative behavior appears to make female song advantageous and female song may in turn enable social interactions that are beneficial to cooperative breeding. The statistical analysis and approach is sound.

I have a few major comments:

1. I recommend tightening and streamlining the discussion to make the valuable and interesting new main points that you make there clear.

Thank you for your careful reading of our manuscript! We have now revised and streamlined the Discussion section.

2. The authors place a lot of weight on directionality / causality of the transition rate results. Be careful not to over extend what can be interpreted from these results. To me, female song enabling transitions to cooperative breeding vs cooperative breeding promoting female song seem likely equally likely possibilities – each of which could occur depending on the circumstances or ancestral traits for specific lineages.

We have now performed additional analyses that allow us to more directly test directionality (phylogenetic path analysis and phylogenetic generalized linear model), and indeed, the evidence supports a bidirectional relationship between cooperative breeding and female song. We now discuss potential mechanisms for evolutionary associations between cooperative breeding and female song in both directions.

3. Similarly, the authors spend a lot of time discussing how female song leads to/enables cooperative breeding and vice versa. Alternatively, they could both be under similar selection pressures (**i.e., there could be a third selective pressure selecting for both**). The authors place a lot of emphasis on the evolution of female song and/or cooperative breeding leading to the other. More attention should be given to this third alternative.

Thanks to this and similar feedback from other reviewers, we now analyze additional factors to test whether the association between female song and cooperative breeding is simply due to a mutual association with an underlying common factor. We found that territoriality, in particular, was strongly associated with both female song and cooperative breeding. We now account for territoriality in all analyses, as well as body mass in the phylogenetic path analyses and phyloglm, and familial living, migration, geographic region (holarctic vs tropical), distance of the range centroid from the equator, size dimorphism, and plumage dimorphism in our phylogenetic generalized linear model analyses. We find that accounting for these factors does not change our finding of a robust association between cooperative breeding and female song.

Specific minor comments:

line 54: odd wording "increased recognition of female song has revealed that this framing may be incomplete". Maybe "increased recognition that female songbirds sing..."

Thank you for pointing out this phrasing! We have made your suggested change (Line 42).

Line 58: Good point. I like the way you state this. See this new reference for the most up to date review of female song function: <https://doi.org/10.1016/j.anbehav.2024.07.018>

Thank you for alerting us to this new paper, which was published after our initial submission. We enjoyed reading it, and we have now cite it in the manuscript (Reference 6, Lines 47-49): "A recent quantitative review reinforces that territory defense, intrasexual competition, and intrapair communication are the most commonly supported functions of female song, while its use in mate attraction appears rare⁶."

Also, see this pre-print for most recent phylogenetic comparative analysis with female song: <https://www.researchsquare.com/article/rs-4018424/v1>. Some of your results complement and others contrast these results. This makes your deep dive into the association of female song and cooperative breeding all the more interesting!

Thank you! This important manuscript is now published, so we have cited the publication in *Nature Communications* instead of the preprint.

Line 65-68: this is a long, complicated sentence but makes important points. Can you break it down to make your points here clearer?

Thank you! We have separated this sentence into two sentences to improve the clarity of the points.

It now reads (Lines 73-77):

"In songbirds, how these complex social systems affect the functions of and selection pressures on male and female song remains poorly understood. However, emerging evidence suggests that there may be important evolutionary interactions between social stability and

vocalizations²¹; for example, cooperative breeding appears to promote the evolution of larger repertoires of functional vocalizations²².”

Line 81-82: "updated our previously published dataset on species-level song features encompassing song complexity and performance metrics for 339 species." I assume this dataset is predominantly male songs? Can you state something about this here or elsewhere. Also, some readers might wonder why you only categorized presence/absence for females. I presume this is because we don't have detailed information of song structure for females in most species. Make sure both of these points are clear somewhere.

Regarding whether the song feature data is from predominantly male songs: We had only stated something to this effect at the end of the introduction (“Here, we compile a dataset of Oscine species that includes species-level cooperative breeding classification, data on female song presence, and several features of (putatively male) song complexity and performance.”), so we added a sentence to the “Song features” section of the Methods that reads “The primary sources of the song-feature data were generally field studies of presumably male birds; when data were specified by sex in the literature, we used only the male song measurements.” Regarding detailed female song categorizations: We now address this directly in the Methods.

line 161: typo - "cooperatively breeding" should be "cooperative breeding"

We apologize for being inconsistent with this wording. Some of the sources we cite use “cooperative breeding” and others use “cooperatively breeding”. We have now tried to consistently use “cooperative breeding” when referring to the classification of cooperative breeding in our dataset and the action of cooperative breeding.

Lines 201-221: I like the rigor with which you tested co-occurrence of traits and that you also looked at transition rate. However, without a formal **path analysis**, I worry about the ability to talk about causality. Be careful not to put too much emphasis on which variable is causal in these transition analyses. I think that some of the transitions, the causal relationship is the other way around. For example, loss of **bi-parental care** is known to lead to the loss of female song (not vice versa). Be careful in your interpretation here. Also, can you make this point clear to the readers?

This section now contains results that, while informative, we agree are not our most conclusive results. We have moved this paragraph to the supplement, and we now present the results of path analyses and phylogenetic generalized linear modeling in the main text as well.

Discussion: Overall, I like your discussion, especially comparing the evidence for sexual competition, social conflict and social cohesion. However, overall I think the discussion could be more streamlined and pointed. I expected three subsequent paragraphs that clearly compared / contrasted evidence for each of these hypotheses. Instead there seems to be several paragraphs that blend into a progressive idea toward arguing for social cohesion. I think your discussion will be more impactful if it more clearly states and compares/contrasts these ideas.

The discussion could also be shorter overall. Condensing the text and arguments will also help make the points clearer and more impactful.

To do this, make each topic and supporting evidence for each concept clearer. Also, social selection, while now being more accepted, is still a newer concept. Be clear what you mean by social competition when you first discuss it.

We have greatly revised the Discussion section, and in particular, have reworked these paragraphs to have clear topic sentences and a parallel structure. Each paragraph begins by explicitly defining the hypothesis and its predictions, presents supporting evidence, evaluates contradicting evidence. We have also streamlined the Discussion overall, removing extraneous text, tightening the arguments, and adding new sections to address reviewer feedback.

Line 252-254: I agree that there is a clear association between female song and cooperative breeding. However, I think it is difficult with your analyses to be certain of the causality of the relationship. I can also easily see how (and it actually seems even more likely to me) that cooperative breeding promotes female song. I suggest removing this causal statement from the first sentence of the discussion.

We have now added path analysis to evaluate the relative strength of models in which cooperative breeding influences female song, and vice versa, with territoriality and body size as additional factors. Indeed, we find statistically equivalent support ($\Delta\text{CICc} < 2$) for models with both directions of influence. We have thoroughly revised the Discussion section, and now avoid making definitive statements about directionality between female song and cooperative breeding.

Line 264: Your analyses and comparisons are within the context of your association of cooperative breeding and female song. Be careful to keep this in the forefront and make sure your readers are aware that your discussion and results are specific to this context and association. For example, the specific statement "However, this predicted trend may be weak in Oscine songbirds" is only true for your associated data of cooperative breeding and female song. In fact, in broader analyses show that intermediate levels of female song is associated with polygynous mating systems. Also, it is not clear what you mean by the predicted trend. Restate this to be clear that this is specifically within the context of cooperative breeding and state what the predictions are.

Thank you for this important clarification. We have revised this paragraph to be more precise about the context of our findings and the expectations under this hypothesis. We now explicitly acknowledge that sexual competition can drive female song evolution in other contexts, while being clear that our analyses specifically address whether sexual competition explains the association between female song and cooperative breeding. This passage now reads (lines 312-316):

"This potential evidence for reduced sexual selection pressure on males in cooperative systems could theoretically correspond to increased pressures on females if cooperative breeding reverses typical sex differences in reproductive variance (per Bateman's principle)¹⁵. However,

empirical evidence from cooperative breeding songbirds suggests that helping is male-biased or mixed-sex but not female-biased³⁴ and female helpers were not typically linked to extreme reproductive skew¹³. Together, these patterns suggest that the conditions necessary for intensified female sexual competition—female-biased helping driven by reproductive monopolization—are not characteristic of cooperative breeding songbirds, casting doubt on the hypothesis that female song evolves in these systems due to reproductive skew.”

Line 274: I did not follow the logic here that this is "consistent with cooperative breeding reducing sexual selection pressures on males and/or increasing them on females." Isn't it consistent with cooperative breeding reducing sexual selection pressures on males and females? Since female song is more stable in monogamous species? I'm not sure I agree that females face more competition in monogamous systems. Also, males often provide direct benefits in polygynous systems like providing food for the mate and offspring. I would think that competition is even between males and females in monogamous systems.

Thank you for pointing out this unclear logic. We have revised this passage to explicitly state the theoretical mechanism connecting reduced male sexual selection to potentially increased female selection, specifically, that cooperative breeding could reverse typical sex differences in reproductive variance (per Bateman's principle). We now cite Hauber and Lacey (2005), who discuss this potential reversal in cooperative breeders. However, we immediately follow this with evidence that this theoretical possibility is not supported by empirical data from cooperative breeding songbirds, which show male-biased or mixed-sex helping rather than female-biased helping patterns. We've removed the largely unnecessary point about female sexual selection in monogamy versus polygyny.

New text, Lines 65-74: "In cooperative systems, sexual selection pressures on females are predicted to increase through intensified intrasexual competition for breeding status and associated increase in variance of reproductive success among females^{11,12}, although the extent of this reproductive skew can vary widely by species, especially in avian taxa¹³. The effects of cooperative breeding on sexual selection pressures in males are even less straightforward^{14,15}. Cooperative breeding often arises from kin-based social structures, with both inclusive and direct fitness benefits for group members, such as resistance to predation through increased group size¹⁶⁻¹⁹. Notably, cooperative breeding often co-occurs with territoriality and stable social bonds²⁰. The fitness benefits of cooperative breeding, therefore, are somewhat entwined with other forms of group living. In songbirds, how these complex social systems affect the functions of and selection pressures on male and female song remains poorly understood.”

Line 276-282: The main points of this paragraph could be clearer.

Thank you for this feedback. We have completely restructured this section to improve clarity. We now begin with a definition of social competition, lay out the predictions of the hypothesis, present supporting evidence that territorial defense is indeed a well-documented function of female song, and discuss our findings that show why this hypothesis only partially explains the cooperative breeding-female song association (Lines 320-337).

Line 283: This is a pivotal paragraph. Clearly state what this paragraph is about at the start (what hypothesis are you investigating / supporting here?).

We now begin this paragraph with a clearer topic sentence (Lines 338-361). It now begins “A third hypothesis is that female song could serve a role in maintaining social cohesion in cooperative breeding groups.”

Line 284-285: I am not sure what you mean by negative and positive interactions. It might not be negative for the bird, if it wins them a territory or mate? Is this the best wording? What about competitive vs affiliative or other neutral language?

We agree that “negative” and “positive” are not the clearest terms to use here and have revised this sentence to use more precise language. The text now reads (Lines 339-342): “Song produced in the context of established breeding pairs is already well-documented as a factor in reinforcing pair bonds; however, evidence suggests that song can also mediate affiliative social interactions beyond the breeding pair, potentially by co-opting the reward systems that support pair-bond maintenance.”

Reviewer #1 (Remarks to the Author):

The referees have all raised aspects of two issues with the original manuscript. The first of these is the degree to which data incompleteness can affect the conclusions, and the second is the ability of comparative methods such as those used to use in this paper to resolve the correlation versus causations problem.

With respect to the first issue I think the authors have done as comprehensive a job as is currently possible to highlight and deal with issues of data unevenness.

With respect to the correlation-causation conundrum, I think there is a range of opinions among comparative biologists on the utility of phylogenetic regressions and path analyses. I have to say that I am reasonably well known as being among the sceptics of the ability of the techniques, but my scepticism has not stopped these techniques being widely implemented and refined. I do not think my scepticism warrants holding up the manuscript any further. I congratulate the author for the comprehensive response to the referees comments

Reviewer #1 (Remarks on code availability):

I have not run the code but it seems that everything is annotated. In response to the comments of all of the referees the authors have provided large numbers of additional analyses in this version. The dataset remains the same. The methodological changes in this version are well portrayed,

Thank you so much for your time and feedback!